# A Review on Mathematical Modeling of Different Biological Methods of Hydrogen Production

**Priyakrishna Yumnam** [1] **and Pradip Debnath** [2,*]

[1] Department of Life Science and Bioinformatics, Assam University, Silchar 788011, India; yumnanpriyakrishna@gmail.com
[2] Department of Applied Science and Humanities, Assam University, Silchar 788011, India
* Correspondence: pradip.debnath@aus.ac.in

**Abstract:** In this paper, we present an updated review on the mathematical modeling of different biological methods of hydrogen production. The presented mathematical modeling and methods range from inception to the current state-of-the-art developments in hydrogen production using biological methods. A comparative study was performed along with indications for future research and shortcomings of earlier research. This review will be helpful for all researchers working on different methods of hydrogen production. However, we only covered biological methods such as biophotolysis, fermentation and microbial electrolysis cells, and this list is not exhaustive of all other methods of hydrogen production.

**Keywords:** biological; biophotolysis; fermentation; microbial electrolysis; mathematical modeling; hydrogen production





## 1. Introduction

Hydrogen has emerged as a clean energy solution with the potential to transform the various sectors of the global economy. As a clean energy carrier, hydrogen has the ability to provide energy without emitting greenhouse gases or pollutants and instead decarbonizes the commercial and industrial sectors [1]. Currently, 74.7% of global electricity is mainly generated from fossil fuels [2]. Fossil fuels and non-renewable energy cause havoc in the soil, water, and air leading to environmental damage and climate change. The partial and complete combustion of fossil fuels emits greenhouse pollutants like COx, NOx, SOx, CxHy, ash, and other organic compounds in the environment [3]. Hydrogen is nontoxic, colorless, odorless [4], tasteless, and the third most abundant element on Earth [5]. On combustion, hydrogen produces water as the end product, which is eco-friendly [6]. The energy content of hydrogen is 122 kJ/g, approximately 2.75 times more than hydrocarbon fuel [6]. Therefore, it is recognized as the cleanest and most promising energy source [7].

With the increasing population and urbanization, the energy demand increased tremendously and is expected to increase by 50% by 2050 [8]. In the future of sustainable resource development, the promising alternative to fossil fuels is the generation of hydrogen fuel from renewable natural resources. About 95% of the hydrogen produced is from natural gas [9]. Some of the general processes used for hydrogen production are stream-forming fossil fuels, partial oxidation of hydrocarbons, photovoltaic-electrolysis system [10], electrolysis [11], coal gasification [12], photoelectrochemical hydrogen production, photoelectrochemical (PEC) [13], and thermolysis [10]. Photoelectrochemical (PEC) hydrogen production is a clean and sustainable method of generating hydrogen gas using sunlight as the primary energy source. It combines principles of both photovoltaics (solar cells) and electrolysis to convert solar energy into chemical energy in the form of hydrogen. When exposed to sunlight, photoelectrolysis, which combines a photovoltaic cell and an electrolyzer, efficiently generates hydrogen by directly converting sunlight into electricity

and initiating the electrolysis process in water [10]. Different ways of producing hydrogen is shown below in Figure 1.

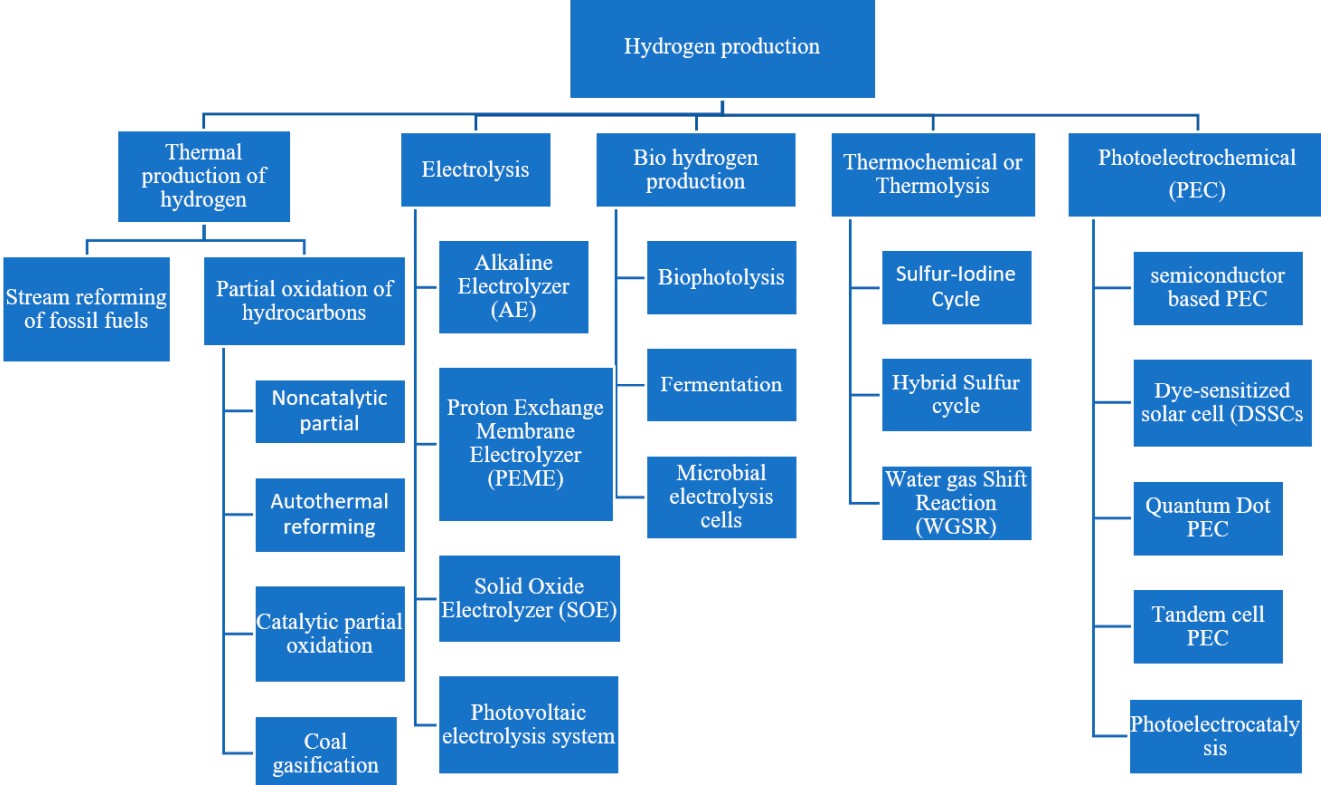

**Figure 1.** Different ways of hydrogen production.

In thermal hydrogen production, hydrogen is generated by the application of heat in the presence of suitable feedstock or reactants. Stream-forming fossil fuels is a catalytic endothermic process at which 970–1100 K is supplied at 3.5 MPa [10]. Meanwhile, partial oxidation of hydrocarbons is an exothermic reaction at moderately high pressure that may or may not require a catalyst [9]. Coal gasification is a recognized process that transforms carbon-based raw materials into synthetic gas by utilizing air, water vapor, or oxygen [12]. Electrolysis is a widely used method for hydrogen production that involves the splitting of water into hydrogen gas and oxygen gas using an electrical current. It is considered a clean and environmentally friendly method because it does not produce greenhouse gases or other pollutants when renewable energy sources are used as the electricity source. The process of electrolyzing, widely used for water splitting, involves immersing a cathode and an anode in an electrolyte, and to improve the current density and reaction rate, catalysts, particularly platinum as a heterogeneous catalyst, are commonly employed [11]. Hydrogen production from thermochemical-based water splitting holds the potential for large-scale production due to its cost-effectiveness and ability to generate significant quantities of hydrogen [13]. Thermolysis of water reduction requires very high temperatures, but using a pair of metal oxides/halides can lower them to below 1000 K [7].

Numerous viable methods for producing hydrogen have been developed, but they suffer from significant drawbacks such as high energy consumption, low conversion efficiency, environmental pollution, and the presence of various impurities in the end products [6]. Hydrogen gas production through biology can be a promising alternative, where room temperature and normal atmospheric pressure, with minimal energy consumption, can produce hydrogen gas. Different types of biological hydrogen production is shown in Figure 2. As the raw materials are waste food crops or organic waste, it is a renewable, waste-reducing, and environmentally clean energy source [14]. Naturally, prokaryotic

cyanobacteria, eukaryotic algae, and photosynthetic purple non-sulfur bacteria (PNSB) produce hydrogen gas [15]. These microbes are used in bio-photolysis under aerobic or anaerobic conditions and fermentation through nitrogen fixation for hydrogen production [16]. Nowadays, electrochemical hydrogen production is infused with microbes to form microbial fuel cells (MFCs) and microbial electrolysis cells (MECs) in the process of developing a more sustainable source of energy [17].

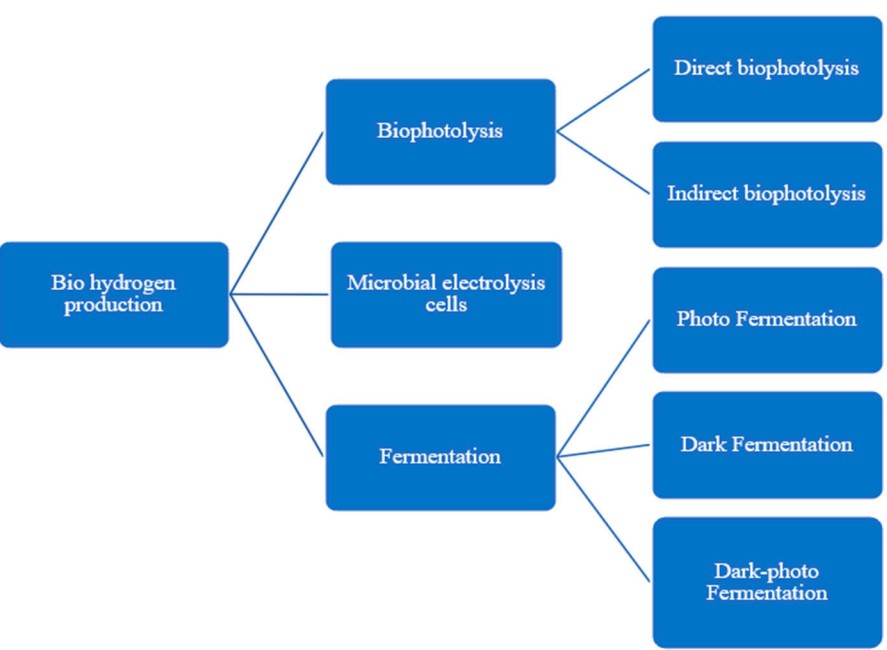

**Figure 2.** Biohydrogen production.

Mathematical modeling describes the biochemical process, kinetics, and interaction involved in the generation of hydrogen gas by microorganisms. Common parameters considered when developing a mathematical model are substrate concentration, growth rate, biomass concentration, reaction rate constant, catalyst property, reaction kinetics, environmental factors (temperature, pH, and pressure), mass transfer coefficient, and energy balance. It involves the components and steps of developing a mass and energy balance equation, incorporating biochemical reaction kinetics and environmental factors (temperature, pH, and pressure), modeling microbial growth, and validating the model with experimental data [18–20]. It involves the selection and consideration of various parameters for describing and predicting the behavior of the system accurately. These parameters may vary depending on the specific hydrogen production method. Additionally, it helped in the understanding of scaling up the model with proper impact assessment.

Hydrogen energy is an efficient and environmentally friendly renewable energy source. Hydrogen can be stored either by physical or chemical methods. Hydrogen can be stored in all states of matter that is solid and gas. Solid hydrogen storage is efficient and suitable for on-board vehicle applications. Solid hydrogen storage is mainly because of physical absorption by van der Waals force or chemical absorption by forming metal hydrides like $MgH_2$, borohydrides, aluminum hydrides, and amino compounds [21]. Hydrogen can be physically stored in the form of compressed gas and liquid and in cryo-compressed form. These different forms of physical storage are achieved by adjusting the pressure for compressed gas storage, manipulating the temperature for liquid storage, or modifying both parameters simultaneously for cryo-compressed gas storage [22]. The reversible storage of simple hydrides poses challenges due to the high temperature, substantial energy input, and slow reaction kinetics typically required [23]. However, storing by adsorption like carbon nano-structuring, metal-organic frameworks (MOFs), covalent organic frameworks (COFs), polymers of intrinsic microporosity (PIMs), and zeolites

provides high hydrogen storage capacity with strong reversibility and rapid kinetics [22]. Nano-structuring increases the surface area and catalytic efficiency; meanwhile, catalyst doping reduces activation energy, enhancing the reaction [21]. $Nb_2O_5$ nanoparticles grafted on a metal-organic framework (MOF) or $Nb_2O_5$@MOF composite significantly improved the hydrogen storage properties of $MgH_2$, including lowering desorption and absorption temperature [24]. Catalysts like Ag0.1Pd0.9/N-ompg-C3N4 for hydrogen production from formic acid enhance the turnover frequency and robust stability [25].

## 2. Biophotolysis

Biophotolysis for hydrogen production is the dissociation or splitting of the water molecule by hydrogenase enzyme under the action of light energy by microorganisms like cyanobacteria and microalgae [26,27].

Biophotolysis utilizes water, sunlight, and $CO_2$ as inputs to produce hydrogen gas, without requiring additional nutrients or substrate. The electrons and protons released after the splitting of water are transferred to chloroplast hydrogenase where they combine forming hydrogen gas [3]. There are three types of hydrogenase, viz. [NiFe] hydrogenases, [FeFe] hydrogenases, and [Fe] hydrogenases [28]. Hydrogenases mainly involved in the biohydrogen production process are [NiFe] hydrogenases and [FeFe] hydrogenases [3]. [Fe] hydrogenases are found in methanogenic archaea bacteria and convert $CO_2$ to $CH_4$ in the presence of $H_2$ [28]. The 4Fe–4S domain, the active site of Fe-Fe hydrogenase, undergoes irreversible inactivation when exposed to oxygen [14]. Only a small amount of algal photosynthetic capacity is used for biohydrogen production. But if the entire capacity of algae for photosynthesis is diverted to hydrogen production, then an acre of land could produce 80 kg of hydrogen [29]. Types of biophotolysis are direct photolysis and indirect photolysis.

### 2.1. Direct Biophotolysis

Photosynthetic microalgae convert water into hydrogen and oxygen utilizing sunlight as the source of energy to break the covalent bond of water.

$$2H_2O + \text{solar radiation} \rightarrow 2H_2 + O_2 \tag{1}$$

$$2H^+ + 2Fd^- \text{ (Ferredoxin)} \rightarrow H_2 + 2Fd \tag{2}$$

In this process, both photosystem I and photosystem II contribute their part in reducing and splitting. PS-I is mainly concerned with the production of reductants for carbon reduction. Meanwhile, in the anaerobic condition, the splitting of water in PS-II generates oxygen and electrons. The light energy absorbed by the photosystem (PS-I and PS-II) transports the electrons to ferredoxin (Fd) from water. This reduced ferredoxin transfers the electrons to the oxygen-sensitive enzyme, hydrogenase, which catalyzes the hydrogen production in the microbes [30,31]. Hydrogenase is both a receiver and donor; the electron that is received from the ferredoxin is donated to a proton converting it into hydrogen gas. In the end, the carbon is reduced in PS-I, leading to the evolution of hydrogen gas with the help of the hydrogenase enzyme. The purity of hydrogen is about 98% which is the highest compared to other mechanisms for biohydrogen production [29].

### 2.2. Indirect Biophotolysis

Indirect biophotolysis is a complex process involving two distinct phases. The first phase is a light-dependent-phase aerobic process, where the microbes synthesize carbohydrates from water and carbondioxide using light energy (Equation (3)). In aerobic conditions with the supply of air, the biomass progressively increases and is concentrated until it reaches a state of stabilization [26,31].

$$6H_2O + 6CO_2 + \text{Light energy} \rightarrow C_6H_{12}O_6 + 6O_2 \text{ (aerobic)} \tag{3}$$

The second phase is the production of hydrogen gas from the synthesized carbohydrates under different metabolic processes. In this, the carbohydrate is broken into simpler fatty acids (Equation (4)), and then these fatty acids are reduced into hydrogen gas (Equation (5)) with the evolution of carbondioxide [32].

$$C_6H_{12}O_6 + 2H_2O \rightarrow 4H_2 + 2CH_3COOH + 2CO_2 \text{ (anaerobic)} \tag{4}$$

$$2CH_3COOH + 4H_2O + Light \rightarrow 8H_2 + 4CO_2 \tag{5}$$

Combining Equations (3)–(5), the overall reaction is

$$12H_2O + light \rightarrow 12H_2 + 6O_2 \tag{6}$$

$$N_2 + 8H^+ + Fd(red)(8H^-) + 16ATP \rightarrow NH_3 + H_2 + Fd(ox) + 16ADP + Pi \tag{7}$$
$$\text{(Hydrogen produced during nitrogen fixation)}$$

$$8H^+ + 8e^- + 16ATP \rightarrow 4H_2 + 16ADP + 16Pi \tag{8}$$
$$\text{(Energy for the reaction)}$$

Non-nitrogen-fixing cyanobacteria such as *Synechococcus*, *Gloeobacter*, and *Synechocystis* are capable of hydrogen production [32]. In filamentous cyanobacteria, the two processes are spatially separated through the formation of a heterocyst. Cyanobacteria like *Oscillatoria*, *Anabaena*, *Calothrix*, and *Nostoc* can fix nitrogen and produce hydrogen. These microorganisms contain nitrogenase and hydrogenase which are highly sensitive to oxygen. These organisms have a modified specialized structure within cyanobacterial cells called heterocysts that aid in nitrogen fixation and hydrogen production. In heterocysts, a microanaerobic condition is maintained. The heterocyst is not only the site for nitrogen fixation, but it also acts as a site for hydrogen production. Hydrogenase not only produces hydrogen but also protect the nitrogenase enzyme from oxygenic attack in carbon-limited condition [33]. This heterocyst contains functional PS I where nitrogenase is located and facilitates the production of hydrogen while fixing nitrogen. Different nitrogenase isoenzymes have varying requirements for paired hydrogen ion fixation [34].

### 2.3. Mathematical Modeling of Biophotolysis

The process of indirect biophotolysis involves sequential stages of aerobic microalgae growth and anaerobic $H_2$ production. The latter stage involves the conversion of carbohydrates into $H_2$ and is primarily anaerobic, occurring when $O_2$ is depleted. The species conservation balance for microalgal species within a well-stirred photobioreactor can be expressed as follows [31]

$$\frac{dy_{algae}}{dt} = y_{algae}\left(y_{algae} - \alpha\right) - \gamma \tag{9}$$

$y_{algae}$ = the microalgae-specific production (growth) rate.
$\alpha$ = is the specific maintenance rate.
$\gamma$ = microalgae biomass consumption while producing $H_2$.

In the anaerobic phase of the cycle in a photobioreactor for microalgae cultivation, the conservation of hydrogen species suggests [31]

$$\frac{dy_{H_2}}{dt} = R_{y_{H_2}/y_{algae}} \cdot \gamma \tag{10}$$

$$\mu_i = \frac{\mu_{max} + I_{av}^n}{\left(\frac{I_{k,max} \cdot I_o}{I_k' + I_o}\right)^n + I_{av}^n} \tag{11}$$

when $i$ = algae or $H_2$.

The species conservation balance for microbial species is [27]

$$V^j \frac{d\, y^j_{algae}}{dt} = \frac{m}{\rho} \left( y^{(j-1)}_{algae} - y^{(j)}_{algae} \right) + V^j y^{(j)}_{algae}(\mu - \alpha) \tag{12}$$

For carbondioxide,

$$V^{(j)} \frac{dy^{(j)}_{CO_2}}{dt} = \frac{m}{\rho} \left( y^{(j-1)}_{CO_2} - y^{(j)}_{CO_2} \right) - V^{(j)} y^{(j)}_{algae} R y_{CO_2/y_{algae}}(\mu - \alpha) \tag{13}$$

For oxygen,

$$V^{(j)} \frac{dy^{(j)}_{O_2}}{dt} = \frac{m}{\rho} \left( y^{(j-1)}_{O_2} - y^{(j)}_{O_2} \right) + V^{(j)} y^{(j)}_{algae} R y_{O_2/y_{algae}}(\mu - \alpha) \tag{14}$$

For hydrogen,

$$V^{(j)} \frac{dy^{(j)}_{H_2}}{dt} = \frac{m}{\rho} \left( y^{(j-1)}_{H_2} - y^{(j)}_{H_2} \right) + V^{(j)} y^{(j)}_{H_2} \left( \mu_{H_2} - \alpha_{H_2} \right) e^{-y_{O_2}/y_{O_{2.sat}}} \tag{15}$$

The correlation between microbial growth (cyanobacteria and algae) and temperature can be understood by [35]

$$\mu_{max} = A_1 \cdot \exp\left( \frac{-E_{a_1}}{RT} \right) - A_2 \cdot \exp\left( \frac{-E_{a_2}}{RT} \right) \tag{16}$$

The influence of radiation and temperature on the growth rate of microbes is denoted by [35]

$$\mu = \frac{\left( A_1 \cdot \exp\left( \frac{-E_{a_1}}{RT} \right) - A_2 \cdot \exp\left( \frac{-E_{a_2}}{RT} \right) \right) \cdot I^n_{av}}{\left( \frac{I_{k,max} \cdot I_o}{I'_k + I_o} \right)^n + I^n_{av}} \tag{17}$$

A hyperbolic function has been suggested to connect growth rate and average light radiation for photo-limited culture which is given by [35]

$$I_{av} = \frac{I_o}{p \cdot C_b \cdot K_a} \left( 1 - \exp\left( -p \cdot C_b \cdot K_a p \cdot C_b \cdot K_a \right) \right) \tag{18}$$

where $I_o$ = external radiance.

$C_b$ = biomass concentration.
$K_a$ = extinction coefficient of biomass.
$p$ = light path.

However, under photosaturation or photoinhibition conditions, the influence of external radiance parameters is considered, and the growth model is given by [35]

$$\mu = \frac{\mu_{max} + I^n_{av}}{I^n_k + I^n_{av}} \tag{19}$$

$$I_k = \frac{I_{k,max} \cdot I_o}{I'_k + I_o} \tag{20}$$

$$\mu = \frac{\mu_{max} + I^n_{av}}{\left( \frac{I_{k,max} \cdot I_o}{I'_k + I_o} \right)^n + I^n_{av}} \tag{21}$$

$\mu$ = specific growth rate.
$\mu_{max}$ = maximum specific growth rate.



$I_k$ = irradiance constant.
$I_{k,max}$ = maximum irradiance constant.
$I_{av}^n$ = average light radiation.

Equation (19) is commonly used when modeling the growth of photosynthetic microorganisms (e.g., algae or cyanobacteria) in biohydrogen production systems. It relates the specific growth rate to the average light intensity and the irradiance constant, which characterizes the sensitivity of the microorganisms to light. Meanwhile, Equation (21) is used in cases where the growth of microorganisms in biohydrogen production is affected not only by light but also by other limiting factors. It takes into account the interaction between average light radiation ($I_{av}$) and the irradiance constants ($I_k'$ and $I_{k,max}$) to calculate the specific growth rate under such conditions.

The Lambert–Beer law describes the relationships of microalgae concentration, light travel distance, and the extinction coefficient [27,31]

$$K_a = Y_p \cdot (b_o - b_1 \cdot C + b_2 \cdot C^2) + Y_b \tag{22}$$

where $b_o$, $b_1$, and $b_2$ are dimensionless coefficients, and $Y_p$ and $Y_b$ ($m^2\ kg^{-1}$) are coefficients, all specific for the microalgae species under consideration.

The average light intensity is calculated using the provided equations [27,36]

$$I_{av} = I_o \cdot e^{-r_{tt} C \cdot K_a} \tag{23}$$

The pH of the medium can be calculated by the following [27]

$$pH = -\log \left( \frac{-k_a + \sqrt{k_a^2 + 4\,k_a[CO_2]}}{2} \right) \tag{24}$$

where $k_a$ = ionization constant.

The maximum temperature at a specific time interval after the minimum temperature is observed is calculated as follows [24]:

$$T_\infty = T_{min} + \frac{\Delta T}{2} + \frac{\Delta T}{2} cos \left[ \frac{\pi(t - t_o)}{t_x} \right] \tag{25}$$

where

$\Delta T = T_{max} - T_{min}$ (change in temperature).
$t$ = simulation time (in seconds).
$t_o$ = initial simulation time.
$t_x$ = time after the minimum temperature occurred (in seconds).

The hydrogen mass production efficiency can be determined by the following [31]:

$$\eta = \frac{y_{H_2\ final}}{y_{algae}(t_{aerobic})} \tag{26}$$

where

$y_{H_2\ final}$ = hydrogen mass fraction at the end of the anaerobic stage.
$y_{algae}(t_{aerobic})$ = total biomass growth (microalgal mass fraction at the end of the aerobic stage).

From Table 1 below, different strains of cyanobacteria are used for hydrogen production. The amount of hydrogen produced by the same strand of cyanobacteria differs because of the environmental factors and the growth medium. *Synechocystis* sp. strain *PCC 6803* accumulates or produces hydrogen at the rate of 143 nmol/mg Chl-a/h and 186 nmol/mg Chl-a/h under the same environmental conditions in different growth mediums [37]. When all the external conditions are similar, *Synechocystis* sp. *PCC 6803*, *Synechococcus* sp. *I12*, *Synechococcus* sp. *I12*, and *Phormidium corium B-26* produce 0.037 μmol $H_2$/mg/h,

0.229 µmol $H_2$/mg/h, 0.019 µmol $H_2$/mg/h, and 0.02 µmol $H_2$/mg/h, respectively [38]. This means that in similar conditions, hydrogen production rates by different organisms are different.

**Table 1.** Hydrogen production by different cyanobacteria.

| Organism | Description | Maximum Hydrogen Production Rate or Accumulation | Growth Condition | Hydrogen Evolution Assay Condition | Hydrogen Enzymes | Ref. |
|---|---|---|---|---|---|---|
| *Cyanothece 51142* | Unicellular and nitrogen-fixing cyanobacteria | 2.13 mL/L/h | ASP2 with nitrogen Temp 30 °C pH 7.4 Light intensity 46 µmol/m²/s | 50 mM glycerol in the medium | Nitrogenase and hydrogenase | [39] |
| *Cyanothece* sp. *Miami BG 043511* | Unicellular nitrogen-fixing cyanobacteria | 16.4 µmol/g dry weight or 15.8 mL/L/h | ASP2 medium, no nitrogen, temp 3 °C, light intensity 30 µEm²/s, diurnal condition | Argon (100%) 30 µEm²/s | Nitrogenase and hydrogenase | [40] |
| *Synechocystis* sp. *PCC 6803* | Non-nitrogen-fixing cyanobacterium | 0.037 µmol $H_2$/mg/h | 70 mL of liquid BG-11 growth medium and aerated 45 µmol photon/m²/s | Argon (100%) 30 µEm²/s | Hydrogenase | [38] |
| *Desertifilum* sp. *IPPAS B-1220* | Filamentous cyanobacterium | 0.229 µmol $H_2$/mg/h | -do- | Argon (100%) 30 µEm²/s | Nitrogenase and hydrogenase | [38] |
| *Synechococcus* sp. *I12* | Thermophilic cyanobacterium | 0.019 µmol $H_2$/mg/h | -do- | Argon (100%) 30 µEm²/s | Hydrogenase | [38] |
| *Phormidium corium B-26* | Filamentous cyanobacterium | 0.02 µmol $H_2$/mg/h | -do- | Argon (100%) 30 µEm²/s | Hydrogenase | [38] |
| *Plectonema boryanum ATCC 18200* | Non-heterocyst nitrogen-fixing cyanobacterium | 0.18 mL/mg/day | Chu #10 growth medium Light intensity 100 µ/m²/s (24 h) pH 7.5, temp 22 °C | Argon and $CO_2$ temp 35 °C Light intensity 100 µ/m²/s | Hydrogenase | [41] |
| *Geitlerinema* sp. *RMK-SH10* | Filamentous cyanobacteria | 0.271 µmol/mg/dry wt/h | ANS III medium Light intensity 30 µmol/m²/s Temp 30 °C | Argon + Medium without Nitrogen + 0.2 M NaCl + 18.9 mmol C-atom/ L glucose + 0.1 µM $Ni^{+2}$, no light, temp 30 °C | Hydrogenase | [42] |
| *Leptolyngbya valderiana BDU 20041* | Nitrogen-fixing cyanobacteria | 0.02 µmol/mg/dry wt/h | ASN III medium Light intensity 13.7 W/m²/s Temp 27–29 °C | Medium without nitrogen No light Temp 27–29 °C | Nitrogenase and hydrogenase | [43] |
| *Synechocystis* sp. *strain PCC 6803* | Mutant formed by disrupting ΔnarB:ΔnirA cyanobacteria | 143 nmol/mg Chl-a/h | BG11 with Nitrogen + 20 mM HEPES Light intensity 40 µmol of photons/m²/s, temp 25 °C, pH 7.5 | Argon in dark for 12 h at room temperature | Nitrogenase and hydrogenase | [37] |
| *Synechocystis* sp. *strain PCC 6803* | Mutant formed by disrupting ΔnarB:ΔnirA cyanobacteria | 186 nmol/mg Chl-a/h | BG11 + 20 mM HEPES Light intensity 40 µmol of photons/m²/s, temp 25 °C, pH 7.5 | Argon in dark for 12 h at room temperature | Hydrogenase | [37] |

Nitrogenase enzyme activation decreases the rate of hydrogen production. The nitrogenase enzyme is actively involved in nitrogen fixation, but in nitrogen-limited conditions, the activity is shifted toward hydrogen production [37]. So, a nitrogen-deficient microenvironment can be created by passing argon, thus creating an anaerobic condition. Apart from biohydrogen production, these microorganisms perform photosynthesis, and therefore, there is a continuous release of oxygen by the system. The oxygen thus decreases the rate

of hydrogen production. So, an anaerobic environment below 0.1% oxygen content must be maintained for continuous and effective hydrogen production [44]. After increasing the temperature by 10 °C from 30 °C, the hydrogen production was twice as much [45].

Table 2 shows that *Selenastrum bibraianum* AARL G052 in a JM-S medium and sulfur-deprived TAP growth medium produced 0.28 μmol/mg Chl-a/h and 0.71 μmol/mg Chl-a/h, respectively, when other conditions remained unchanged [46]. A genetically modified *Cyanothece ΔhupL* mutant and *Cyanothece C ΔhupL* mutant produced 84.0 ± 21.6 μmol H$_2$/mgChl-a/day and 2224.8 ± 434.4 μmol H$_2$/mgChl-a/day. But wild-type *Cyanothece* PCC7822 produces 2474.4 ± 496.8 μmol H$_2$/mgChl-a/day [33]. *Chlorellaceae pyrenoidosa* in a TAP medium and TCP medium produce 23.12 mL/L and 93.86 mL/L mL/L, respectively [47]. *C. protothecoides* produce 59.5 mL/L and 82.5 mL/L when the hydrogen-evolving medium is changed [48]. When any of the investigating factors is changed or altered, it affects the rate of hydrogen production.

**Table 2.** Hydrogen production by different algae.

| Organism Strains | pH | Temp | Light Intensity | Growth Medium | Hydrogen Evolution Assay Condition | Hydrogen Production Rate or Accumulation | Ref. |
|---|---|---|---|---|---|---|---|
| *Tetraselmis subcordiformis* | 7.6 | - | - | BG-11 growth medium with sulfur | medium + deionized water and chloride compounds instead of sulfur | 1.73 ± 0.31 cm$^3$/h | [18] |
| *Selenastrum bibraianum* AARL G052 | - | 25 °C | 30.8 μmol photon/m$^2$/s (24 h) | JM-S medium | - | 0.28 μmol/mg Chl-a/h | [46] |
| *Selenastrum bibraianum* AARL G052 | - | 25 °C | 30.8 μmol photon/m$^2$/s (24 h) | Sulfur-deprived TAP | - | 0.71 μmol/mg Chl-a/h | [46] |
| *Chlamydomonas reinhardtii (CC425)* | 7.2 | 24 °C | 60 μmol photon/m$^2$/s 12 h light/12 h dark | TAP + air | Sulfur-deprived TAP | 17.02 ± 3.83 μmol/L/h | [16] |
| *Chlamydomonas moewusii (SAG 24.91)* | 7.2 | 24 °C | 60 μmol photon 12 h light/12 h dark/m$^2$/s | TAP + air | Sulfur-deprived TAP | 5.12 ± 0.37 μmol/L/h | [16] |
| *Cyanothece* PCC7822 | - | 30 °C | Continuous light at 30–50 μmol photon/m$^2$/s | BG-11 aerated | Medium with no nitrogen No oxygen | 2474.4 ± 496.8 μmol H$_2$/mg Chl-a/day | [33] |
| *Cyanothece ΔhupL* mutant | - | 30 °C | Continuous light at 30–50 μmol photon/m$^2$/s | BG-11 | Medium with no nitrogen No oxygen | 84.0 ± 21.6 μmol H$_2$/mg Chl-a/day | [33] |
| *Cyanothece C ΔhupL* | - | 30 °C | Continuous light at 30–50 μmol photon/m$^2$/s | BG-11 | Medium with no nitrogen No oxygen | 2224.8 ± 434.4 μmol H$_2$/mg Chl-a/day | [33] |
| Desmodesmus armatus var. bicaudatus AARL G019 | - | 25 °C | 30.8 μmol photon/m$^2$/s (24 h) | JM-S | - | 0.30 μmol/mg Chl-a/h | [46] |
| *Desmodesmus armatus var. bicaudatus AARL G019* | | 25 °C | 30.8 μmol photon/m$^2$/s (24 h) | TPA-S | - | 0.15 μmol/mg Chl-a/h | [46] |
| *Chlamydomonas reinhardtii 137c* | 7.2 | 25 °C | 100 μmol photon/m$^2$/s | TAP without sulfur | TAP + sulfur + nitrogen | 2.5 mL/L/h | [49] |
| Marine *C. Pyrenoidosa* IOAC707S | 7.2 | 28 °C | 25 μmol photon/m$^2$/s (24 h) | TAP | TAP-P + 30 g/L NaCl | 22 mL/L | [50] |

**Table 2.** *Cont.*

| Organism Strains | pH | Temp | Light Intensity | Growth Medium | Hydrogen Evolution Assay Condition | Hydrogen Production Rate or Accumulation | Ref. |
|---|---|---|---|---|---|---|---|
| *Stigeoclonium* sp. AARL G030 | - | 25 °C | 30.8 μmol photon/m²/s (24 h) | TPA-S | - | 0.27 μmol/mg Chl-a/h | [46] |
| Marine *C. Pyrenoidosa* IOAC707S | 7.2 | 28 °C | 25 μmol photon/m²/s (24 h) | TAP | TAP-P and sea water | 38 mL/L | [50] |
| *Chlamydomonas reinhardtii* CC124 | 7.2 | 28 °C | 70 μmol photon/m²/s | TAP medium | Sulfur-deprived TAP | 3.3 mL/L/h | [51] |
| *Chlorellaceae pyrenoidosa* | 7 | 28 °C | 180 μmol photon/m²/s (24 h) | TAP medium | Sulfur-deprived TAP (alternate day dark: light) | 23.12 mL/L | [47] |
| *Chlorellaceae pyrenoidosa* | 7 | 28 °C | 180 μmol photon/m²/s (24 h) | TCP medium | TCP + DCMU (alternate day dark: light) | 93.86 mL/L | [47] |
| *Chlorella* sp. AARL G014 | - | 25 °C | 30.8 μmol photon/m²/s (24 h) | JM-S medium | - | 0.46 μmol/mg Chl-a/h | [46] |
| *Chlorella* sp. AARL G014 | - | 25 °C | 30.8 μmol photon/m²/s (24 h) | Sulfur-deprived TAP | - | 0.49 μmol/mg Chl-a/h | [46] |
| *C. protothecoides* | 7.3 | 30 °C | 30–35 μmol photon/m²/s (14 h light: 10 h dark) | TPA + 0.35 mM N₄Cl | Nitrogen-deficient TPA 24 h light | 59.5 mL/L | [48] |
| *C. protothecoides* | 7.3 | 30 °C | 30–35 μmol photon/m²/s (14 h light: 10 h dark) | TPA + 0.35 mM N₄Cl | Nitrogen- and sulfur-deficient TPA 24 h light | 82.5 mL/L | [48] |

The above Tables 1 and 2 show that cyanobacteria can produce hydrogen at lower light intensity than microalgae. Cyanobacteria showed less energy demand than microalgae. In the case of marine microalgae, marine species showed the least energy demand for biohydrogen production. Marine microalgae *C. Pyrenoidosa* IOAC707S can actively produce hydrogen when light intensity is as low as 25 μmol photon/m²/s [50]. Microalgae that have the [Fe-Fe] hydrogenase enzyme have an efficiency of 12–14% in converting solar radiation into hydrogen [52]. In most of the above cases, the favorable temperature and pH are more or less similar. The growth medium and the hydrogen evolution assay condition may or may not be the same. The addition or removal of other substances in the growth medium or hydrogen evolution assay medium may alter the enzymatic activities of the microorganism which affect the hydrogen production rate. In the biophotolytic process, the optimum temperature for microbial growth ranges from 20–35 °C. In the case of thermophilic cyanobacterium, the optimum temperature differs greatly, and it rises to 55 °C [26].

## 3. Fermentation

Fermentation is a biological process of converting organic substrates, such as waste materials or renewable feedstocks, into hydrogen-rich biogas by microorganisms. It involves a complex interplay of microbial interactions and biochemical reactions, where a complex organic molecule is degraded into simpler substances. Anaerobic microorganisms perform dark fermentation from organic waste materials generating VFAs and other soluble metabolic products along with $CO_2$ and $H_2$. These VFAs are actively consumed by PNSB in the presence of sunlight, releasing $H_2$ as the byproduct [53].

### 3.1. Dark Fermentation

Dark fermentation or heterotrophic fermentation occurs in an oxygen-free environment as many species are sensitive to oxygen. Since there is no utilization of light, it can be performed at any time, and the hydrogen yield is high [3]. Complex organic materi-

als, such as sugars, starches, cellulose, and organic waste, serve as substrates for dark fermentation [54]. These materials are broken down into simpler compounds through various enzymatic reactions. The hydrogen gas is released as an intermediate byproduct of first-stage breakdown [3]. During dark fermentation, the hexose sugar is broken down into VFAs like acetic acid, butyric acid, propanoic acid, and ethanol. When glucose is broken down into acetic acid and propanoic acid, there is no yield of hydrogen gas. But oxygen can inhibit the activity of hydrogen-producing bacteria and redirect their metabolic pathways toward less desirable byproducts, reducing the efficiency of hydrogen production [54–57].

$$C_6H_{12}O_6 + 2H_2O \rightarrow 2C_2H_4O_2 + 2CO_2 + 4H_2 \ \Delta G_0 = -206 \text{ kJ} \tag{27}$$

$$C_6H_{12}O_6 + 2H_2O \rightarrow C_4H_8O_2 + 2CO_2 + 2H_2 \ \Delta G_0 = -254 \text{ kJ} \tag{28}$$

$$3C_6H_{12}O_6 + 2H_2O \rightarrow C_2H_4O_2 + C_3H_6O_2 + 2CO_2 + 2H_2O \tag{29}$$

These intermediate products can either be consumed by purple sulfur nitrogen bacteria (PSNB) for photofermentation or enter other metabolic processes. But in the second stage, many methanogenic bacteria utilized hydrogen as an electron donor for anaerobic fermentation [29]. Microorganisms, through normal glycolytic pathways, convert glucose into pyruvic acid, thereby releasing ATP and NADH. Pyruvate is fermented into various byproducts, depending on the microorganisms and conditions. Through the acetic acid fermentation and butyric acid fermentation process, pyruvate is further converted into acetic acid and butyric acid [57,58].

$$C_6H_{12}O_6 \rightarrow 2C_3H_4O_3 \text{ (pyruvate)} + ATP + NADH \tag{30}$$

$$C_3H_4O_3 \text{ (pyruvate)} \rightarrow CH_3COOH \text{ (acetic acid)} + CO_2 + NADH \tag{31}$$

$$2C_3H_4O_3 \text{ (pyruvate)} \rightarrow C_4H_8O_2 \text{ (butyric acid)} + CO_2 + NADH \tag{32}$$

Naturally occurring waste products are the raw material for dark fermentation, and the byproducts of dark fermentation are clean energy sources like hydrogen gas, economically important substances like alcohols, and VFAs, which are again raw materials for photofermentation. So, the efficiency of hydrogen production and the type and quantity of byproducts depend on factors such as substrate composition, pH, temperature, and reactor design. Mo, Ni, EDTA, yeast extract, and ethanol are both positive and negative regulators, while methanol, Cu ions, and sulfide ions show only positive regulation; meanwhile, Fe, vitamins, buffer solutions, Mg, and NaCl show negative regulation [59]. The effects of the aforesaid conditions will be mathematically shown in the later part of this article.

### 3.2. Photofermentation

Photofermentation is a simple process where a photosynthetic microorganism has the ability to convert solar energy into chemical energy in the form of hydrogen bonds from an organic substrate acting as a carbon source. Photofermentation is carried out by purple non-sulfur bacteria (PNSB), a group of anaerobic facultative microorganisms like *Rhodobacte*, *Rhodopseudomonas*, and *Rhodospirillum* sps. In the presence of light and the absence of molecular oxygen, PNSB have the ability to convert diverse organic substrates into molecular hydrogen [60]. However, PNSB have only one fixed intracellular photosystem. The electrons generated during the reduction of carbon create a potential difference leading to the pumping of protons across the membrane. In this process, ATP is generated, which aids the electron movement through a substantial electron carrier until the final electron receptor ferredoxin. With no nitrogen gas, ATP from the surroundings reduced the proton, utilizing the same electron from ferredoxin, thereby producing hydrogen gas [29]. This is a

high-energy-requiring process yet highly productive as every single proton can be reduced to hydrogen gas. Hydrogen production in the absence of nitrogen is as follows [61]:

$$8e^- + 8H^+ + 16ATP \rightarrow 4H_2 + 16ADP + 16Pi \tag{33}$$

In the presence of nitrogen, the metabolic pathway differs. The nitrogenase enzyme takes the hydrogen to form ammonia. Here, a small amount of hydrogen is released as a byproduct. Nitrogen is highly sensitive to oxygen and causes irreversible change in the active site if it comes in contact with oxygen [61].

$$N_2 + 8H^+ + 8e^- + 16ATP \rightarrow 2NH_3 + H_2 + 16ADP + 16Pi \tag{34}$$

The production of hydrogen from different substrates during photofermentation is as follows [60]:

$$\text{Glucose}: C_6H_{12}O_6 + 2H_2O + \text{Light} \rightarrow 12H_2 + 6CO_2 \ \Delta H_R^O = 360.884 \ kJ/mol \tag{35}$$

$$\text{Acetate}: C_2H_4O_2 + 2H_2O + \text{Light} \rightarrow 4H_2 + 2CO_2 \ \Delta H_R^O = 129.428 \ kJ/mol \tag{36}$$

$$\text{Butyrate}: C_4H_8O_2 + 2H_2O + \text{Light} \rightarrow 10H_2 + 4CO_2 \ \Delta H_R^O = 352.684 \ kJ/mol \tag{37}$$

$$\text{Lactate}: C_3H_6O_3 + 3H_2O + \text{Light} \rightarrow 6H_2 + 3CO_2 \ \Delta H_R^O = 231.942 kJ/mol \tag{38}$$

This process derives energy from organic waste and simultaneously generates valuable compounds. Conditions such as optimum temperature, pH, light intensity, other enzymes or compounds, and reactor designs are important factors for hydrogen production.

### 3.3. Dark Photofermentation

Dark photofermentation is a simultaneous realization of dark and photofermentation in a single reactor tank, producing hydrogen gas as byproducts. Here, the organic compounds from various sources are degraded into VFAs by the process of dark fermentation, and then PNSB use this dark fermentation effluent as a substrate for photofermentation. The simplest chemical reaction can be represented by the following [62].

During dark fermentation when glucose is used as an initial substrate,

$$C_6H_{12}O_6 + 2H_2O \rightarrow 2CH_3COOH + 4H_2O + CO_2 \ \Delta G^0 = 104.6 \times 2 = 209.2 \ kJ \tag{39}$$

During photofermentation when acetic acid is the carbon source,

$$2CH_3COOH + 4H_2O \rightarrow 8H_2 + 4CO_2 \ \Delta G^0 = 104.6 \times 2 = 209.2 \ kJ \tag{40}$$

Simultaneously occurring dark and photofermentation can be summarized as

$$C_6H_{12}O_6 + 6H_2O \rightarrow 12H_2 + 6CO_2 \ \Delta G^0 = 3.2 \ kJ \tag{41}$$

The theoretical yield of the dark photofermentative yield is always lower than the theoretically calculated value. Yet, the overall yield is comparatively higher when the fermentation process is performed individually.

Despite the advancement in research efforts, there is still a dearth of studies conducted at pilot and industrial scales. To address this gap, the utilization of mathematical models proves invaluable. These models enable the simulation of the impact of different environmental and operational variables on the process, aiding in process control and design for scaling-up purposes. Presently, there have been few review papers discussing mathematical models of fermentative hydrogen production. This review aims to fill that gap and provide

insights into the potential applications of fermentation as well as to guide future research in this specialized domain.

*3.4. Mathematical Modeling of Dark Fermentation and Photofermentation*

The effective and proper functioning of the fermentation depends on the substrate concentration, the medium of nutrients, and inhibitors present in the medium, along with environmental factors like pH, temperature, and the reactor. The mathematical model demonstrates the relationship between the kinetic growth of the microorganisms and the hydrogen production rate and the environmental conditions. There are different types of operational designs that support different types of growth models, degradation of the substrate, and hydrogen production. The mathematical relations are shown below:

Substrate uptake rate, assuming substrate $S_1$ in a substrate-limited condition, is described by Monod's equation [19].

$$\{\frac{dS_1}{dt}\}_u = -k_1 \left( \frac{S_1}{K_{S1} + S_1} \right) X_1 \tag{42}$$

where, $k_1$ is the maximum substrate consumption rate, and $K_{S1}$ is the half-saturation constant.

The growth of microorganisms described by the Monod equation analyzed the growth of species with limited availability of a single substrate as a food source, and the organism consumed the substrate for growth and reproduction without any inhibitory or toxic effects [19].

The change in the alkalinity or acidity in the environment affects the substrate utilization by the microbes [19]. Therefore, the above Monod's equation, Equation (42) can be written as

$$\{\frac{dy}{dx}\}_u = -k_1 \left( \frac{S_1}{K_{S1} + S_1} \right) X_1 I \tag{43}$$

where $I$ is the pH inhibition term.

With the change in pH, substrate uptake kinetics is changed in both the upper and lower ends of pH [63], and I can be represented as

$$I = \{e^{\left(-3\left(\frac{pH - pH_{UL}}{pH_{UL} - pH_{LL}}\right)^2\right)}, \text{ if } pH < pH_{UL} \, ; \, 1, \text{ if } pH > pH_{UL}\} \tag{44}$$

where $pH_{UL}$ is the upper limit for pH inhibition, and $pH_{LL}$ is the lower limit for pH inhibition; at these upper and lower points, the microorganism's action is prohibited completely.

Considering biomass growth is proportional to the biomass yield ($Y_1$), chemical oxygen demand uptake rate, and cell death ($k_{d1}$) [19], the net biomass growth can be determined by

$$\frac{dX_1}{dt} = Y_1 \left(\frac{dS_1}{dt}\right)_u - k_{d1} X_1 \tag{45}$$

Or, with a limited supply of nutrients, high biomass concentration, or any change in the environmental temperature or light intensity, the microbes show a logistic growth curve [64] where the cell concentration is plotted against time.

$$X = \frac{X_{max}}{[1 + \exp(-k_c \cdot t)\left(\frac{X_{max}}{X_o} - 1\right)]} \tag{46}$$

where $X$ is the cell concentration, $t$ is time, $k_c$ is the apparent specific growth rate, $X_o$ is the initial cell concentration, and $X_{max}$ is the maximum cell concentration.

The mother substrate is broken down into a simpler substrate that is volatile fatty acids (VFAs) $S_2$, $S_3$, and $S_4$ [19]. The reaction rates can be described by the equations

$$\frac{dS_2}{dt} = (1 - Y_1) f_{1-2} \left(\frac{dS_1}{dt}\right)_u \tag{47}$$

$$\frac{dS_3}{dt} = (1 - Y_1)f_{1-3} \left(\frac{dS_1}{dt}\right)_u \tag{48}$$

$$\frac{dS_4}{dt} = (1 - Y_1)f_{1-4} \left(\frac{dS_1}{dt}\right)_u \tag{49}$$

It can be generalized as

$$\frac{dS_n}{dt} = (1 - Y_1)f_{i-j} \left(\frac{dS_1}{dt}\right)_u \tag{50}$$

where $f_{i-j}$ are the stoichiometric coefficients of the fermentation product, $j$ is associated with the consumption of $i$, and $I \& j = 1, 2, 3. \ldots$

Hydrogen production can be obtained from the equation

$$\frac{dS_H}{dt} = (1 - Y_1)f_{1-5} \left(\frac{dS_1}{dt}\right)_u + (1 - Y_2)f_{2-5} \left(\frac{dS_2}{dt}\right) + (1 - Y_3)f_{3-5} \left(\frac{dS_3}{dt}\right) + (1 - Y_4)f_{4-5} \left(\frac{dS_4}{dt}\right) \tag{51}$$

In the Gompertz equation, the dynamics of cultured microbial growth is under a steady condition where the rate of substrate utilization equals the rate of substrate input [65]. It simplified the complex interaction within microbial communities. Hydrogen production rate, according to the Gompertz equation, is given by

$$P_{H_2}(t) = H_{max} \cdot \exp\left\{-\exp\left[\frac{R_{max}\,e}{H_{max}}(\lambda - t) + 1\right]\right\} \tag{52}$$

where $P_{H_2}(t)$ is the hydrogen accumulation in time $t$, $H_{max}$ is the maximum cumulative hydrogen, $R_{max}$ is the maximum hydrogen production rate, $\lambda$ is the lag phase, and $e$ is Eulero's number.

Assuming a single microbial culture's growth and metabolic activity are representative of the entire system in a continuous condition, where the microbial culture is continuously fed with a substrate and is simultaneously harvesting the hydrogen, by employing the Luedeking–Piret equation, the correlation of bacterial growth and biohydrogen production can be deducted from (Equation (53)) [20]. It helped to quantify how the growth of these cells and the efficient generation of biohydrogen were interconnected under ideal conditions.

$$\frac{1}{X}\frac{dP_i}{dt} = \alpha_i \frac{1}{X}\frac{dX}{dt} + \beta_i \tag{53}$$

where $\frac{1}{X}\frac{dX}{dt}$ is the specific growth rate ($\mu$), and $\frac{1}{X}\frac{dP_i}{dt}$ is the specific hydrogen production rate ($v$). The values of $\alpha_i$ and $\beta_i$ are the intercept and the slope of $\frac{1}{X}\frac{dX}{dt}$ vs. $\frac{1}{X}\frac{dP_i}{dt}$ of the exponential phase.

Volumetric hydrogen production rate (VHP):

$$\text{VHP} = \frac{V_{hydrogen}}{V_{BR} * t} \tag{54}$$

where $V_{hydrogen}$ is the volume of H$_2$ produced, VBR is the working volume of the bioreactor, and $t$ is the time of H$_2$ production.

$$\text{VHP} = \beta_o + \beta_1 X_1 + \beta_2 X_2 + \beta_3 X_3 + \beta_{11} X_1^2 + \beta_{22}\beta_2^2 + \beta_{33}X_3^2 + \beta_{12}X_1X_2 + \beta_{13}X_1X_3 + \beta_{23}X_2X_3 \tag{55}$$

where $b_1$ to $b_3$ are linear coefficients; $b_{11}$, $b_{22}$, and $b_{33}$ are quadratic coefficients; and $b_{12}$, $b_{13}$, and $b_{23}$ are interaction coefficients.

All the analytical determinations were performed in triplicate, and average results are presented [65].

$$Y = \frac{C * V}{m} \times 100 \tag{56}$$

where $Y$ is the reducing sugar yield. $C$ is the reducing sugar concentration, $V$ is the fermentation liquor volume, and $m$ is the initial dry matter weight of raw material.

### 3.5. Mathematical Modeling of Dark Photofermentation

In dark photofermentation, microorganisms responsible for dark fermentation and photofermentation are simultaneously cultured together in the same medium [53]. Here, the microorganism responsible for dark fermentation ($X_1$) breaks down the initial substrate ($C_g$) into volatile fatty acids ($C_{Ac}, C_{Bu}$) which are substrates for photofermentative bacteria ($X_2$).

The biomass of dark fermentative bacteria ($X_1$):

$$\frac{dC_{X_1}}{dt} \; \mu_1 * C_{X_1} \tag{57}$$

The biomass of photofermentative bacteria ($X_2$):

$$\frac{dC_{X_2}}{dt} = \mu_2 * C_{X_2} \tag{58}$$

where $\mu_2$ = the sum of the substrates of photofermentation (volatile fatty acids), i.e., $\mu_{Ac} + \mu_{Bu}$. The concentration of the initial substrate:

$$\frac{dC_g}{dt} = \frac{1}{Y_{C_{X_1}/C_g}} \mu_1 C_{X_1} \tag{59}$$

where $Y_{C_{X_1}/C_g}$ = biomass yield coefficient with respect to the initial substrate.

The concentration of a volatile fatty acid (acetic acid) as a substrate for photofermentation and product for the dark fermentation:

$$\frac{dC_{Ac}}{dt} = -\frac{1}{Y_{C_{X_2}/C_{Ac}}} \mu_3 C_{X_2} + Y_{C_{Ac}/C_{X_1}} * \mu_1 C_{X_1} \tag{60}$$

where

$Y_{C_{X_2}/C_{Ac}}$ = biomass yield coefficient with respect to acetic acid.

$Y_{C_{Ac}/C_{X_1}}$ = product (acetic acid) yield coefficient in terms of biomass.

The concentration of a volatile fatty acid (butyric acid) as a substrate for photofermentation and product for the dark fermentation:

$$\frac{dC_{Bu}}{dt} = -\frac{1}{Y_{C_{X_2}/C_{Bu}}} \mu_4 C_{X_2} + Y_{C_{Bu}/C_{X_1}} * \mu_1 C_{X_1} \tag{61}$$

where

$Y_{C_{X_2}/C_{Bu}}$ = biomass yield coefficient with respect to butyric acid.

$Y_{C_{Bu}/C_{X_1}}$ = product (butyric acid) yield coefficient in terms of biomass.

Production of hydrogen:

$$\frac{dV_{H_2}}{dt} = \left( \left( \frac{R * T}{P_t} \right) * (V_l) * \left( \frac{Y_{H_2}/C_{X_1} * \mu_1 C_{X_1}}{2} \right) + \left( \frac{Y_{H_2}/C_{X_{2_{Ac}}} * \mu_3 * C_{X_2}}{2} \right) + \left( \frac{Y_{H_2}/C_{X_{2_{Bu}}} * \mu_4 * C_{X_2}}{2} \right) \right) - \left( ((K_L a)_{H_2} * \left( \frac{K_{H_2} * V_{H_2} * P_t}{R * T * V_g} - C_{H_2} \right) ) \right) \tag{62}$$

where

$Y_{H_2}/C_{X_1}$ = product (hydrogen) yield coefficient in terms of biomass (initial substrate).
$Y_{H_2}/C_{X_{2_{Ac}}}$ = product (hydrogen) yield coefficient in terms of biomass (acetic acid).
$Y_{H_2}/C_{X_{2_{Bu}}}$ = product (hydrogen) yield coefficient in terms of biomass (butyric acid).
$(K_L a)_{H_2}$ = the mass transfer coefficient of $H_2$.

$K_{H_2}$ is Henry's constant.

$$\frac{dC_{H_2}}{dt} = ((K_L\,a)_{H_2} * (\frac{K_{H_2} * V_{H_2} * P_t}{R * T * V_g} - C_{H_2}))) \tag{63}$$

Energy conversion efficiency based on carbon conversion, $ECE_{CC}$ [53]

$$ECE_{CC}(\%) = \frac{Mole\ of\ hydrogen\ produced * HHV\ of\ hydrogen}{Moles\ of\ substrate\ consumed * HHV\ of\ substrate} * 100 \tag{64}$$

where HHV is higher heating values.

Overall energy conversion efficiency, $ECE_o$ [53]

$$ECE_o(\%) = \frac{Mole\ of\ hydrogen\ produced * HHV\ of\ hydrogen}{Moles\ of\ substrate\ consumed * HHV\ of\ substrate + I * A * t} * 100 \tag{65}$$

where $I$ = light intensity, $A$ = illuminating areas of both sides of the FPPBR, and $t$ = time of operation.

The energy conversion efficiency analysis of biohydrogen production through fermentation can be determined by the following [65]

$$E = \frac{V_{H_2} \times Q_{H_2}}{Q_{heat} \times m} \times 100\% \tag{66}$$

where $V_{H_2}$ is the volume of hydrogen gas, $Q_{H_2}$ is hydrogen heating value, $Q_{heat}$ is the heating value of substrate stover, and $m$ is the dry matter weight of the substrate.

*Rhodobacter*, *Rhodopseudomonas*, *Clostridium*, and *Enterobacter* sp. are widely used in the fermentation process for biohydrogen production (Table 3). In most of the cases, only one microorganism is used for the study. So, there are few comparative studies on the amount of hydrogen produced by a mixed cultured system. Different types of substrates including wastewater, seed sludge, cheese whey, rice husk, VFAs like lactic acid and malic acid, and carbohydrates like glucose, lactose, and sucrose are used. For experimental studies, pure organic substrates like glucose, sucrose, or VFAs are used, but for large and industrial-scale production of biohydrogen, different types of waste products (wastewater, farm waste) are used [66,67]. The same substrate can be utilized by different organisms for hydrogen production in the fermentative process (Tables 3–5). The addition of other substrates elevates hydrogen production [68,69]. pH is an important determining factor, as some acidogenic microbes prefer to multiply at lower pH while some prefer healthy growth under normal conditions. Therefore, the pH range for the optimum multiplication of microorganisms depends on the type of microorganism, and it ranges from 5 to 8. Clostridium species showed optimum growth at 5–6, most of the species of *Enterobacter*, *Rhodobacter*, *Rhodopseudomonas*, and most *Chlamydomonas* at normal pH around 6 to 7.5, while *Chlamydomonas* MGA 161 and *Rhodobacter sphaeroides* favor alkalinity of 8 to 8.5 [61,68–72]. At specific temperatures, the metabolic pathway of the microbes is altered. The microorganisms can perform at normal temperatures [73,74]. In the case of dark fermentation, light is not required, but for photofermentation and dark photofermentation, light is mandatory. The light intensity for different organisms varies, and there is always a saturation point after which the organism shows a decrease in hydrogen production. Usually, the optimum intensity for photofermentative bacteria lies between 4000 and 5000 lux [75,76]. *Rhodobacter sphaeroides* showed the highest hydrogen production rate of 41.74 mL/L/h and 0.012 L/L/h with the biohydrogen production of 960 mL $H_2$/L malate and 0.008 L/L/h inoculum when malate and malate with a nutrient supplement were used [68,69]. In the same environmental condition of the dark fermentative process, the mixed culture of *Clostridium* and *Enterobacter* has the highest hydrogen production rate of 2608 mL $H_2$/L/day and 3793 mL $H_2$/L/day when substrates are rice husk and dilute

acid hydrolysis of rice husk + 0.75 mg cellulase/mL [77]. In this case, the addition of other additive nutrient mediums or enzymes increases the total biohydrogen production.

**Table 3.** Fermentative biohydrogen production by microorganisms.

| Microorganism | Fermentative Process | Substrate | Biohydrogen Production | Highest Hydrogen Production Rate | Environmental Factors | Ref. |
|---|---|---|---|---|---|---|
| *Rhodobacter capsulatus DSM 1710.* | Photofermentation | Acetic acid | 0.05 to 0.11 g $H_2$/g acetete | 1.04 mmol/L/h | Light intensity 263.6 W/m$^2$ Acetate concentratiom 35.35 mM VSS (suspended volatile substance) 0.27 gVSSL/L | [78] |
| *Rhodobacter sphaeroides* | Photofermentation | Mallic acid | 960 mL $H_2$/L malate | 41.74 mL/L/h | pH (6.5 to 8) Light intensity (35–185 W/m$^2$) Carbon/nitrogen (15–35) | [68] |
| *Rhodopseudomonas palustris* | Photofermentation | Dark fermented palm oil mill effluent (DPOME) | 30.59 mL. $H_2$/g-COD$_{removed}$ | 0.514 mL/h | pH-6 Inoculum substrate percentage 20% Light intensity 350 W/m$^2$ | [79] |
| *Rhodobacter sphaeroides* | Dark and photofermentation | Malic acid + 0.2 g/L yeast extract | 0.008 L/L/h | 0.012 L/L/h | Light intensity (150–250 W/m$^2$) pH 7 to 8.25 | [69] |
| *R. capsulatus* | Photofermentation | Dark fermentation effluent + lactose + glucose | 1.59 mmol $H_2$/mL$_{medium}$ | 208.4 mmol $H_2$/Ld | Temperature (35 ± 3 °C) Light intensity (70 photon μmole/m$^2$/s) agitation (130 rpm) | [74] |
| *Enterobacter aerogenes* | Dark fermentation | Carbohydrate in cheese whey (CW) | 32.5 g/L CW | 24.7 mL/L/h | Temperature (25–37 °C) pH (5.5–7.5) | [73] |
| *Clostridium and Enterobacter* | Dark fermentation | Rice husk | 320.6 mL/g DAH | 2608 mLH$_2$ /L/day | pH of 7 to 7.5 | [77] |
| *Clostridium and Enterobacter* | Dark fermentation | Dilute acid hydrolysis of rice husk + 0.75 mg cellulase/mL | 473.1 mL/g DAH | 3793 mL $H_2$ /L/day | pH of 7 to 7.5 | [77] |

The kinetic model offers a comprehensive understanding of the rate of chemical reactions enabling optimization of hydrogen production by adjusting reaction conditions like temperature, pH, catalysts, and reactor design. This model aids in identifying bottlenecks and inefficiencies of hydrogen production. By providing insight dynamics of the microbial population, rate-limiting steps, and inefficiencies within the process, it helps to select and manipulate specific microbial strains to improve hydrogen production. In addition, it can be used to evaluate the suitability of different organic substrates for biohydrogen production and the prediction of expected yield under specific conditions [64,80,81].

From Table 4, it can be seen that different types of organisms exhibit different forms of substrate consumption rates. In some cases, zero-order and first-order kinetics are also seen [64,69,82]. The unstructured and non-segregated Monod substrate consumption model fits many microbial substrate consumption models [20,73,80,81,83]. The linear regression between the saturation constant and the maximum specific growth rate helps in determining hydrogen production with time [84]. When the nutrient or carbon supply is ample, the microbial growth expands exponentially; then, when it reaches a point where the nutrient or carbon source becomes exhausted, there is a decline in the growth rate. This growth pattern is a logistic model. This growth model consists of three distinct phases: the log phase (the initial stage of biomass accumulation which is slow but progressive), the exponential phase (multiple division with the highest accumulation of biomass), and the stationary phase. It has a sigmoidal curve graph when represented. In certain cases, with a mixed population of microorganisms or in an unstructured system, the Monod growth model is also seen [81–83]. To calculate the cumulative hydrogen production on

the reactors, the Gompertz model or the modified Gompertz model is used; meanwhile, the Leudeking–Piret model is preferred in the case of hydrogen production by increasing microbial growth.

**Table 4.** Kinetic model of fermentation process.

| Microorganism | Substrate | Biomass Growth Rate Model | Substrate Consumption Model | Model of Hydrogen Production | Ref. |
|---|---|---|---|---|---|
| *Mix culture* | Seed sludge | Monod | Monod | - | [83] |
| *Rhodobacter capsulatus DSM 1710* | Acetic acid and lactic acid | Logistic model | Lactic acid—first-order kinetics Acetic acid—zero and first order | Modified Gompertz | [64] |
| *Mixed population* | Wastewater | Monod | First-order kinetics Michaelis_Menten-based | Monod | [82] |
| *Rhodopseudomonas palustris* | Malic acid, glutamic acid, and FeCl$_3$ | Logistic | Monod | Leudeking–Piret | [80] |
| *R. sphaeroides O.U.001* | Cheese whey effluent from dark fermentation | Logistic | Monod | Modified Gompertz equation and Luedeking–Piret model | [42] |
| *Rhodobacter sphaeroides* | Malic acid (C) (photo and dark) | Logistic | First order | - | [69] |
| *Clostridium pasteurianum (dark) Rhodopseudomonas palustrisWP3–5* | Sucrose from effluent of dark fermentation | - | - | Modified Gompertz equation | [85] |
| *Rhodobacter sphaeroides* | Glucose and acetic acid (dark and photo) | Monod and logistic | Monod | Modified Gompertz equation | [81] |
| *Enterobacter aerogenes* | Cheese whey | Logistic equation | Monod | Modified Gompertz equation | [73] |

A parametric model is a mathematical representation that relates various process parameters to the performance of the production system. The process is used for process optimization by adjusting parameters such as catalysts, temperature, pressure, and flow rates. This model emphasized the study of sensitivity analysis where variation in specific parameters affects the overall process performance and understanding of the robustness of the system. Furthermore, it helped in selecting appropriate equipment sizes, materials, and configurations based on desired performance and constraints [86–88].

A parametric model is frequently used for determining the effects of various conditions that affect hydrogen production at one time. Most of the time, the Box–Behnken design (BBD), the Plackett–Burman design (PBD), and the central composite design (CCD) are used for investigation. Using a full factorial design, all possible combinations of the factors that may aid in hydrogen production were tested. Table 5 shows the investigation of the effect of malic acid, glutamate, and FeCl$_3$ on hydrogen production by *Rhodopseudomonas palustris* [80]. It shows the level of significance of interactions among the investigating factors. *Rhodobacter sphaeroides*, *Rhodobacter sphaeroides DSM 158*, and *Enterobacter aerogenes 2822* followed BBD which combined elements of two level-factorial and incomplete block design within a three-level strategy. The effect of the temperature and pH on hydrogen production by *Enterobacter aerogenes 2822* was studied using BBD [73]. A two-level factorial design of Plackett–Burman was used to investigate the effects of the concentration of organic acids, temperature, and light intensity on hydrogen production from cheese whey effluent by *R. sphaeroides O.U.001* [71]. Li et al., [87] used a combination of Plackette–Burman

and central composite design (five-level factorial design) to investigate the effects of pH, temperature, and inoculation amount in hydrogen production by *HAU-M1.*

**Table 5.** Parametric model used for fermentation.

| Microorganism | Substrate | Method | Investigating Factor | Ref. |
|---|---|---|---|---|
| *Rhodopseudomonas palustis WP3–5* | Acetate (C) and glutamate (N) | Ordinary differential equations | Biomass growth, acetic acid, COD, PHB | [88] |
| *Rhodobacter capsulatus DSM 1710.* | Acetic acid (C), sodium glutamate (N) | The RSM and Box–Behnken design | Initial substrate, initial VSS, and light intensity | [78] |
| *R. sphaeroides O.U.001* | Cheese whey effluent | Plackett–Burman | Organic acid concentration, temperature, and light intensity | [20] |
| *Rhodopseudomonas palustris* | Malic acid, glutamic acid, and FeCl$_3$ | Full factorial design | Malic acid, glutamate, FeCl3 | [80] |
| *HAU-M1* | Platanus orientalis leaves | Plackett–Burman and central composite design (CCD) | pH, temperature, and inoculation amount | [87] |
| *Rhodobacter sphaeroides* | Sewage water | Box–Behnken design and central composite design | Difference in the concentration of carbon and nitrogen source | [86] |
| *R. palustris* | Immobilized inoculum from dark fermented palm oil mill effluent (DPOME) | RMS | pH, inoculum-substrate percentage (ISP), and light intensity | [79] |
| *Rhodobacter sphaeroides DSM 158* | Malic acid and glutamic acid | RSM and Box–Behnken experimental design (BBD) | pH, carbon-to-nitrogen ratio, and light intensity | [68] |
| *Enterobacter aerogenes 2822* | Cheese whey | Box–Behnken | Temperature, pH | [73] |

## 4. Microbial Electrolysis Cells

Microbial electrolysis cells or electrochemical-assisted microbial reactors use the integrated technology of electrolysis of hydrogen (electrohydrogenesis) assisted by microbes for direct conversion of biodegradable organic materials into hydrogen [89]. In microbial fuel cells, electrodes are suspended in a nutrient medium containing different types of organisms like fermentative, acetoclastic methanogenic, and electricigenic microorganisms in the anodic chamber and hydrogenotrophic methanogenic microorganisms in the cathodic chamber. In the anodic chamber, with the help of a small amount of electricity, the microorganism degrades the waste organic material, releasing electrons. These electrons pass through the perforated biofilm, heading toward the cathodic chamber. Here, with the help of hydrogenotrophic methanogenic microorganisms and a small potential difference (<1.23 V), the electrons are excited to pair with a proton (H$^+$), forming a molecule of hydrogen [90,91]. The cathodic chamber is strictly maintained with no oxygen supply as bacterial growth flourishes only in anaerobic conditions. The current in the electrodes boosts the growth of the microorganisms. The change in or fluctuation of potential difference across the electrode influences the rate of hydrogen production [90].

The mathematical modeling of MECs is to acknowledge the importance and effects of different factors like the concentration of organism biomass, carbon source, amount of potential difference, etc., in hydrogen production. There are three major domains of microbial electrolysis cell analysis by using a mathematical model. They are as follows [90].

The simplified biofilm growth model: Major analysis related to chemical reaction kinetics; mainly concerned with the concentration profile of the microorganism, carbon source, and hydrogen production.

The time-dependent growth model: Analysis of dynamic biofilm parameters; mainly concerned with the distribution and concentration of substrates and anodic microorganisms.

The production time space growth model: It is related to geometrical separation and electrical parameters for reactor design. The distribution of current density and electrode potential is based on conductivity and the nature of fluid flow.

### 4.1. Simplified Biofilm Growth Model

The hydrogen production rate in a simplified biofilm growth model can be determined as follows [92]:

$$Q_{H_2} = Y_{H_2} \left( \frac{I_{MEC}}{mF} \frac{RT}{P} \right) - Y_h \mu_h x_h V \tag{67}$$

The hydrogen production rate can be determined as follows [93]:

$$Q_{H_2} = Y_{H_2} \, A_a \frac{I_{MEC}}{mF} \frac{RT}{P} \tag{68}$$

$Y_{H_2}$ is cathode efficiency, $Y_h$ is the yield rate for hydrogen, $R$ is the universal gas constant, $P$ is pressure in the anode compartment, $T$ is temperature in MECs, and $A_a$ is the anode area.

In the outer anodic layer, where there is a continuous flow of wastewater in a microbial fuel cell (MFC), where the effluent and influent are the same, and wastewater is the carbon source [92]:

$$\frac{dS}{dt} = -q_f x_f + D \, (S_o - S) \tag{69}$$

where

$x_f$ is the fermentative microorganism.

$q_f$ is the substrate consumption rate by the fermentative microorganism.

$S_o$ and $S$ are the organic substrate concentration in the influent and the anodic compartment.

In a simplified MEC model, in which two biofilms are used, acetate is used as the substrate; then [92]

$$\frac{dA}{dt} = -q_e x_e - q_m(x_{m,1} + x_{m,2}) + D \, (A_o - A) + Y_{COD} q_f x_f \tag{70}$$

where

$x_e$ is the concentration of electricigenic microorganisms.

$x_f$ is the concentration of fermentative microorganisms.

$x_m$ is the acetoclastic methanogenic microorganism.

$q_e$ is the substrate consumption rates by electricigenic microorganisms.

$q_m$ is the substrate consumption rates by acetoclastic methanogenic microorganism.

$q_f$ is the substrate consumption rate by fermentative microorganism.

$D$ is the dilution rate.

$Y_{COD}$ is acetate yield from an organic substrate.

$A_o$ and $A$ are the acetate concentration in the influent and the anodic compartment.

When utilizing wastewater or acetate as a carbon source, there is growth in the microbial population [92].

The growth rate of fermentative microorganisms in the outer anodic layer is

$$\mu_f = \mu_{max,f} \frac{S}{K_{S,f} + S} \tag{71}$$

where $\mu_{max}$ is the maximum growth rate, and $K$ is the Monod half rate constant.

The growth rate of acetoclastic methanogenic microorganisms in the outer and inner anodic layers is [92]

$$\mu_m = \mu_{max,m} \frac{A}{K_{A,m} + A} \tag{72}$$

The growth rate of electricigenic microorganisms in the biofilm layer is [92]

$$\mu_e = \mu_{max,e} \frac{A}{K_{A,e} + A} + \frac{M_{ox}}{K_M + M_{ox}} \tag{73}$$

According to Pinto [92] the consumption of substrate by different types of microorganisms can be denoted as:

For fermentative microorganisms,

$$q_f = q_{max,f} \frac{S}{K_{S,f} + S} \tag{74}$$

where $q_{max}$ is the maximum consumption rate.

For acetoclastic methanogenic microorganisms in the outer and inner anodic layers,

$$q_m = q_{max,m} \frac{A}{K_{A,m} + A} \tag{75}$$

For electricigenic microorganisms in the biofilm layer,

$$q_e = q_{max,e} \frac{A}{K_{A,e} + A} + \frac{M_{ox}}{K_M + M_{ox}} \tag{76}$$

The biomass of the microorganism can be determined by the mass balance equation [92]. The mass balance for fermentative microorganisms in outer anodic biofilm layer 1 is

$$\frac{dx_f}{dt} = \mu_f x_f - K_{d,f} x_f - \alpha_1 x_f \tag{77}$$

The mass balance for acetoclastic methanogenic microorganisms in outer and inner anodic layers 1 and 2 is

$$\frac{dx_{m,1}}{dt} = \mu_m x_{m,1} - K_{d,m} x_{m,1} - \alpha_1 x_{m,1} \tag{78}$$

$$\frac{dx_{m,2}}{dt} = \mu_m x_{m,2} - K_{d,m} x_{m,2} - \alpha_2 x_{m,2} \tag{79}$$

The mass balance for electricigenic microorganisms in inner anodic biofilm layer 2 is

$$\frac{dx_e}{dt} = \mu_e x_e - K_{d,e} x_e - \alpha_2 x_e \tag{80}$$

The mass balance for hydrogenotrophic methanogenic microorganisms in cathodic layer 3 is

$$\frac{dx_h}{dt} = \mu_h x_h - K_{d,h} x_h - \alpha_3 x_h \tag{81}$$

The biofilm retention constant $\alpha$ is denoted as [92]

$$\alpha_k = \begin{cases} \sum \frac{(\mu_\lambda x_\lambda - K_{d,\lambda} x_\lambda)}{\sum x_\lambda}, & if \ (\sum x_\lambda)_k \geq X_{max,k} \\ 0, & othrewise \end{cases} \tag{82}$$

where $X_{max,k}$ is the maximum attainable biomass concentration in the $k$-th layer, and $k = 1,2,3. \ldots x_\lambda$ is the population in the $k$-th layer.

For layer 1, $\lambda = f, m_1$, for layer 2, $\lambda = e, m_2$, and for layer 3, $\lambda = h$.

$\mu$ is the growth rate in the *k*-th layer.

The biofilm retention constant $\alpha$ is denoted as [93]

$$\alpha = \begin{cases} \frac{\mu_a X_a + \mu_m X_m}{X_a + X_m}, & if\ X_a + X_m > X_{max} \\ 0, & otherwise \end{cases} \tag{83}$$

where

$\mu_a$ and $\mu_m$ are the growth rates.

$X_a$ & $X_m$ are the concentrations of anodophilic and methanogenic microorganisms.

The biofilm retention constant $\alpha$ for biofilms 1 and 2 is [93]

$$\alpha(X_1,\ X_2\ ;\ X_{max}) = \frac{1}{2}D[1 + \tanh(K_{\varkappa}(X_1 + X_2 - X_{max}))] \tag{84}$$

where $K_{\varkappa}$ is curve steepness for biofilm retention.

The intracellular electron transfer from the anode (oxidized mediator fraction) to the cathode (reduced mediator) can be represented as [92]:

$$M_{Total} = M_{red} + M_{ox}$$

where

$$M_{ox} = -Y_m q_e \frac{\gamma}{V x_e} \frac{I_{MEC}}{mF} \tag{85}$$

$\gamma$ is the mediator molar mass.

$m$ is the number of electrons transferred per mol of the mediator.

$Y_M$ is the oxidized mediator yield.

$M_{ox}$ is the oxidized mediator fraction per electricigenic microorganism.

$M_{red}$ is the reduced mediator fraction per electricigenic microorganism.

$F$ is the Faraday constant.

$I_{MEC}$ is current in MECs.

The rate of oxidized mediator fraction per electricigenic microorganism [94]:

$$\frac{dM_{ox}}{dt} = -Y_M\ q_e(S, M_{ox})X_e + \frac{\gamma}{VmF}I \tag{86}$$

$S$ is substrate.

MEC voltage can be calculated by [92]

$$-E_{applied} = E_{CEF} - \eta_{ohm} - \eta_{conc} - \eta_{act} \tag{87}$$

where $\eta_{ohm}$ is the ohmic overpotential, $\eta_{conc}$ is the concentration overpotential, and $\eta_{act}$ is the activation overpotential.

$$\eta_{conc} = \eta_{conc,A} + \eta_{conc,c} \tag{88}$$

$\eta_{conc,c} = 0$.

$$\eta_{ohm} = I_{MEC}\ R_{int} \tag{89}$$

$R_{int}$ = is internal resistance.

MEC current can be calculated by [92]

$$I_{MEC} = \frac{E_{CEF} + E_{applied} - \frac{RT}{mF}\ln\left(\frac{M_{total}}{M_{red}}\right) - \eta_{act,C}}{R_{int}} \tag{90}$$

where

$$R_{int} = R_{min} + (R_{max} - R_{min})e^{-K_R X_e} \tag{91}$$

$R_{min}$ is the lowest internal resistance observed.

$R_{min}$ is the maximum internal resistance observed.

$K$ is the constant.

MEC current generated by anodophilic microorganisms can be calculated by [93]

$$I_{MEC} = \left( \gamma_S k_a \mu_a X_a L_f (1 - f_S^0) + \gamma_X b X_a L_f \right) A_{sur} \tag{92}$$

where $\gamma_S$ and $\gamma_X$ are the yield coefficients, $k_a$ is the yield coefficient for anodophilic microorganisms, $\mu_a$ is the growth rate of anodophilic microorganisms, $X_a$ is the concentration of anodophilic microorganisms, $L_f$ is the biofilm thickness, $f_S^0$ is the electron fraction, $b$ is the decay coefficient, and $A_{sur}$ is the anode surface area.

### 4.2. Time-Dependent Growth Model

The Haldane kinetics for the rate of substrate consumption by methanogenic microorganisms is [93]

$$\mu_a(S) = \mu_{a,max} \frac{S}{S + K_{S,a} + \frac{S^2}{K_1}} \tag{93}$$

where $\mu_{a,max}$ is the maximum specific growth rate, $K_{S,a}$ is the maximum half-life rate concentration of $S$, and $K_1$ is the inhibition constant.

The mass balance distribution of substances in a steady state is [93]

$$D_e \frac{d^2 S^{bio}}{dz^2} - \rho_x [\mu_a(S^{bio})\phi_a + \mu_e(S^{bio}, E_a^{bio})\phi_e] = 0 \tag{94}$$

At a constant volume of the anodic chamber, the mass balance for the substrate is [93]

$$\frac{dy}{dx} = \frac{F}{V_a}[S^{in} - S] - \frac{A_a}{V_a}[\overline{\mu_a}(S^{bio})L_f \rho_x \phi_a + \overline{\mu_e}(S^{bio}, E_a^{bio})L_f \rho_x \phi_e] \tag{95}$$

where

$S^{bio}$ is the substrate concentration.

$\rho_x$ is the biofilm density.

$\phi_a$ is the mass fractions of acetoclastic methanogenic microorganisms.

$E_a^{bio}$ is potential variation through the biofilm.

The biomass of acetoclastic methanogenic microorganisms is [93]

$$\frac{d}{dt}(A_a x_a) = A_a k_1 \overline{\mu_e}(S^{bio})x_a - A_a b_{in} x_a - A_a r_{det} \phi_a \tag{96}$$

The biomass of electricigenic microorganisms is [93]

$$\frac{d}{dt}(A_a x_e) = A_a a k_4 \overline{\mu_e}(S^{bio}, E_a^{bio})x_e - A_a[b_{in} + \overline{r_{res}}(E_a^{bio})]x_e - A_a r_{det} \phi_e \tag{97}$$

The potential loss due to electron flow across the biofilm is [93]

$$\frac{dE_a^{bio}}{dt} = \frac{dj_z}{dz} + \frac{F}{\gamma}[k_5 \mu_e(S^{bio}, E_a^{bio}) + k_7 r_{res}(E_a^{bio})]\rho_x \phi_e \tag{98}$$

Potential variation at a steady state is [93]

$$k_{bio} \frac{d^2 E_a^{bio}}{dz^2} - \frac{F}{\gamma}[k_5 \mu_e(S^{bio}, E_a^{bio}) + k_7 r_{res}(E_a^{bio})]\rho_x \phi_e = 0 \tag{99}$$

The expected current when all the electrons reach the anode can be calculate by [93]

$$I = \frac{F}{\gamma} A_a [k_5 \overline{\mu_e} (S, E_a) + k_7 r_{res}(E_a)] x_e \tag{100}$$

where $K_5$ and $K_7$ are constant.

The current density is represented as [95]

$$j = j_{max} \frac{S}{K_{s,app} + S} \tag{101}$$

where

$j_{max}$ is the maximum current density.
$S$ is the substrate concentration.
$K_{s,app}$ is the apparent half-saturation substrate concentration in the biofilm.

The Nerst–Monod equation to calculate current density in a biofilm is [95]

$$j = -j_{max} \left( \frac{1}{1 + exp \left[ \frac{F}{RT} (E - E_{KA}) \right]} \right) \tag{102}$$

where $E_{KA}$ is the potential difference when $j = 1/2\ j_{max}$.

$$j_{max} = \gamma_s q_{max} X_f L_{fa} \tag{103}$$

$\gamma_s$ is a conversion factor.
$q_{max}$ is the maximum specific rate of substrate utilization.
$X_f$ is the active biomass concentration in the biofilm.
$L_{fa}$ is the biofilm thickness.

The Butler–Volmer equation for anodic current loss in the electrode interface is [95]

$$j = -j_o \exp \frac{nF (1 - \alpha)(E_{anode} - E^o_{interface})}{RT} \tag{104}$$

where

$j_o$ is the current density.
$\alpha$ is the electron transfer coefficient for anodic and cathodic reactions.
$E_{anode}$ is the anode potential.
$E^o_{interface}$ is the standard potential.

The hydrogen flow rate or production rate:

$$qH_2 = \epsilon_{cat} \frac{\gamma k_{H_2}}{F} I \tag{105}$$

where $\epsilon_{cat}$ is the cathode efficiency.

*4.3. Production Time-Space Growth Model*

Biohydrogen production at the cathode can be calculated using the model's current density result and Faraday's constant [96]

$$V_H = \frac{22.4 A_c}{zA} \int_0^t J_{c,av} dt \tag{106}$$

In the context of modeling biohydrogen production, the assumptions regarding current density typically involve simplifications such as assuming steady-state conditions,

ideal electrode behavior, and simplified kinetics to facilitate computational efficiency and model tractability.

According to the Nernst–Monod relationship, the microbial activity, substrate availability, and transfer coefficients are essential factors influencing the electrochemical reactions occurring at the anode [97]. The electrochemical activity occurring at the anode surface by the hydrogen-producing microorganisms leads to the transfer of electrons to the anode surface, producing electric current and the generation of biohydrogen, which is represented as

$$j_{bioan} = nFq_{max}X_{bf}Z_{bf}\left(\frac{C_i}{C_i + K_i}\right)\left(\frac{1}{1 + \exp\left(\frac{-F}{RT}(\varphi_1 - \varphi_2 - E_{\frac{1}{2}})\right)}\right) \tag{107}$$

where

$j_{bioan}$ is the current density in the anode.
$X_{bf}$ is the density of active biomass.
$Z_{bf}$ is the thickness of the biofilm in the anode.
$C_i$ is the concentration of the electrogenic substrate.
$K_i$ is the half-max-rate substrate concentration.
$\varphi_1$ is the electrode potential.
$\varphi_2$ is the potential of the biofilm.
$E_{\frac{1}{2}}$ is the anode potential with the half-maximum rate of electrogenic substrate consumption.

The electrochemical kinetics at the anode is given by [96]

$$J_a = J_0\left[\frac{M_{red}X_e}{M_{red}X_e}\exp\left(\frac{(1-\alpha)F}{RT}E_a - \varphi_a - E_{eq.a}\right) - \left[\frac{M_{ox}X_e}{M_{ox}X_e}\exp\left(\frac{-\alpha F}{RT}E_a - \varphi_a - E_{eq,a}\right)\right.\tag{108}$$

where

$J_0$ is the exchange current density.
$E_a$ is the anode potential.
$E_a$ is the anode potential.
$E_{eq,a}$ is the equilibrium potential of the anode reaction.
$\varphi_a$ is the electrolyte potential, where $\alpha$ is the transfer coefficient.
$X_e$ is the electroactive microbial biomass.
$M_{red}$ is the reduced species of the mediator.
$M_{ox}$ is the mediator to its oxidized form.

Assuming that the rate-determining step occurs at the electrode surface and there is a linear relationship between the reaction rate and overpotential in the homogenous environment of a continuous steady state, the electrochemical kinetics at the cathode by the Tafel equation is [96]

$$J_c = J_{o,c} \times 10^{(E_c - \varphi_c - E_{eq,c})/b} \tag{109}$$

where

$J_{o,c}$ is the exchange current density.
$E_c$ is the cathode potential.
$\varphi_c$ is the electrolyte potential close to the cathode.
$E_{eq,c}$ is the equilibrium potential of the cathode reaction.
$b$ is the Tafel slope.

The strength of the current depends on the reaction rates of the anode and cathode and the medium conductivity and interfacial electric potential ($\varphi_a$ and $\varphi_c$) determined by overpotential [96]. The current continuity equation between the electrodes, where variation in $\varphi_a$ and $\varphi_c$ is established:

$$\nabla\cdot J = -\nabla\cdot\sigma\nabla\varphi \tag{110}$$

where

$J$ is the current density vector.
$\sigma$ is the conductivity.
$\varphi$ is the electrolyte potential.

The potential difference in nonuniform reaction rates can be determined by [97]

$$-k_{bio}\left(\frac{1}{r}\frac{\delta}{\delta r}r\frac{\delta\varphi_2}{\delta r} + \frac{\delta^2\varphi_2}{\delta z^2}\right) = \Gamma_{bio}j_{max}\left[\frac{1}{1 + \exp(\frac{-F}{RT}(\varphi_1 - \varphi_2 - E_{1/2}))}\right] - a_{elec}i_{oc}\exp(\frac{-\alpha_c F}{RT}(\varphi_1 - \varphi_2 - E_{ocp})) \qquad (111)$$

where

$\frac{\delta}{\delta r}$ and $\frac{\delta^2}{\delta z^2}$ are the partial derivatives with respect to spatial coordinates $r$ and $z$.
$i_{oc}$ is the exchanged current density.
$\Gamma_{bio}$ is the specific surface area of the developed attached biofilm.
$a_{elec}$ is the specific area of the porous matrix.
$j_{max}$ is the maximum current density.
$E_{ocp}$ is the open circuit potential.
$\alpha_c$ is the charge transfer coefficient.

After analyzing the mass balance in the reactor with flowing liquid, Equation (112) considers the influence of both axial and radial flow depression coefficients, which helps in understanding the reactor's hydraulic characteristic impact on the overall process [97]

$$D_{ax}\frac{d^2C_i}{dz^2} + D_r\left(\frac{1}{r}\frac{d}{dr}r\frac{dC_i}{dr}\right) - \frac{U_{eff}dC_i}{\epsilon dz} - R_i = \frac{dC_i}{dt} \qquad (112)$$

where

$D_{ax}$ and $D_r$ are the axial and radial dispersion coefficients.
$U_{eff}$ is the linear velocity.
$C_i$ is the substrate concentration.
$\epsilon$ is the apparent electrode porosity.
$\left(\frac{1}{r}\frac{d}{dr}r\frac{dC_i}{dr}\right)$ is the radial dispersion term.
$\frac{dC_i}{\epsilon dz}$ is the convection term.

The mass transport model in the biofilm is [96]

$$\frac{dS}{dt} + \nabla\cdot\left(-D_{S,eff}\nabla S\right) = R_S W_S = -q_e x_e \qquad (113)$$

where

$D_{S,eff}$ is the effective diffusion coefficient.
$R_S$ is the substrate consumption rate.
$W_S$ is the substrate molecular weight.
$x_e$ is the concentration of electroactive microorganisms.
$q_e$ is the substrate consumption rate per mass of electroactive microorganisms.

The mass balance in the liquid phase with a direct conductive mechanism is given by [96]

$$\frac{dS_l}{dt} + \nabla\cdot(-D_{S,l}\nabla S_l) + u\cdot\nabla S_l = R_{S,l}W_S = -q_e x_e \qquad (114)$$

where

The subscript $l$ refers to the liquid phase.
$D_{S,l}$ is the diffusion coefficient.
$D_{S,l}\nabla S_l$ is the diffusive flux.

The mass balance of the electricigenic microorganism is [96]

$$\frac{dX_e}{dt} = (\mu_e - K_{d,e} - \alpha_e)x_e \tag{115}$$

where

$x_e$ is the electroactive biomass concentration.
$K_{d,e}$ is the microbial decay rate.
$\alpha_e$ is the biofilm retention constant.
$\mu_e$ is the growth rate per electroactive biomass.

The mass balance in the recirculation tank can be identified by using [96,97]

$$\frac{dCe_i}{dt} = \frac{Q}{V}(C_{outi} - Ce_i) \tag{116}$$

$$\frac{dS_T}{dt} = \frac{Q}{V_T}(S_R - S_T) - R_{S,T}W_S \tag{117}$$

where

$Q$ is the volumetric flow rate.
$V$ is the volume of the tank.
$C_{outi}$ is the concentration of the microorganism at the outlet.
$Ce_i$ is the concentration of the microorganism in the reactor.
$S_T$ is the substrate concentration in the recirculation tank.
$S_R$ is the average substrate concentration at the MEC outlet.
$R_{S,T}$ is the substrate consumption rate in the tank.

$$S = \frac{\frac{(V_{H,2} - V_{H,1})}{V_{H,1}}}{\frac{(I_2 - I_1)}{I_1}} \tag{118}$$

where

$(I_2 - I_1)$ is the variation of input (conductivity or flow rate).
$I_1$ is the base.

Sensitivity, as defined by Equation (118), quantifies how a change in an input parameter (such as conductivity or flow rate) affects the output (the generated hydrogen volume) [96]. In this context, it helps us understand how changes in conductivity and flow rate impact the overall hydrogen production process.

### 4.4. Optimizing Features Affecting the MEC System Design for Hydrogen Production

4.4.1. Anode

Carbon-based materials, like graphite and carbon fibers, are commonly used due to their conductivity and microbial adhesion properties. Recycled carbon fiber anodes can offer cost-effective improvements [98,99]. Increasing the surface area of the anode, using porous materials of carbon, and simplifying the reaction improves biofilm growth and current density. But carbon-based materials have high intrinsic ohmic resistance leading to high energy loss at a large scale [100,101]. Titanium wire and Molybdenum are used for corrosion resistance and durability. Stainless steel, while less biocompatible, has good conductivity and potential for hydrogen generation [98].

4.4.2. Cathode

The choice of cathode material in hydrogen and methane production processes is critical. Methane production requires a lower voltage ($-0.23$ V vs. SHE), but hydrogen production occurs at a higher voltage ($-0.41$ V vs. SHE) [102]. Steel and nickel are common non-precious metal options due to their cost-effectiveness, conductivity, and resistance

to corrosion. Stainless steel, especially when it has a large specific surface area, can rival platinum for hydrogen production. Nickel, on the other hand, offers high corrosion resistance and efficient electron transfer. These materials are widely used in pilot-scale systems, with stainless steel being a cost-effective favorite [98]. Methanogenic activity increases when temperature rises above 35 °C [103].

### 4.4.3. pH

The pH level in microbial electrolysis cells (MECs) significantly impacts their performance. Differences in pH between the anode and cathode can lead to high overpotentials. Low cathode and high anode pH improved hydrogen production [104]. Most MEC studies are conducted at a pH of 7 due to neutral microbial activity [105]. Weak acids in the electrolyte can improve MEC characteristics by increasing conductivity and reducing impedance. Phosphate is commonly used as an electrolyte in MECs due to its positive effects on hydrogen generation and current density [98].

### 4.4.4. Temperature

Temperature plays a crucial role in microbial electrolysis cells (MECs). One set of electromethanogenesis studies took place at room temperature (approximately 22–25 °C), while the remaining research focused on mesophilic conditions, which involve slightly higher temperatures (around 30–35 °C) [100]. Most microbes thrive at 35–40 °C, enhancing substrate degradation and power generation. A temperature of 31°C is considered an efficient operating temperature for MECs based on COD removal and microbe loading. However, some MECs can produce hydrogen at lower temperatures, like 4 °C, while minimizing methane generation [98].

### 4.4.5. Applied Potential Difference

According to the voltage applied in the MEC, the ratio of methane and hydrogen gas production changes. In an experiment by Nam et al. [106], it was observed that a potential of 0.2 V produces hydrogen at the maximum; meanwhile, methane production was decreased. Adjusting this voltage significantly affects the growth of electroactive bacteria (EAB) and methane generation [100]. To produce hydrogen efficiently at the MEC cathode, a minimum of 0.2 V is needed to overcome the thermodynamic barrier [98].

## 5. Advantages and Disadvantages of Different Types of Biohydrogen Production

Biohydrogen production is a promising avenue in renewable energy research. Different types of biohydrogen production methods or processes offer unique benefits and challenges in terms of efficiency, sustainability, and scalability. Analyzing the different processes, some of the advantages and disadvantages are summarized in Table 6.

**Table 6.** Advantages and disadvantages of different types of processes used for biohydrogen production.

| Mode of Operation | Advantages/Benefits | Disadvantages/Limitations |
|---|---|---|
| Direct biophotolysis | 1. Hydrogen is produced from water in the presence of sunlight. <br> 2. There is no additional substrate or nutrients required for the process. <br> 3. ATP or energy is not required in the process. <br> 4. $CO_2$ is decreased in the environment. <br> 5. Hydrogen that is 98% pure can be obtained. | 1. $O_2$ inhibits the production of hydrogen. <br> 2. Hydrogenase enzyme is sensitive to oxygen. <br> 3. Productivity of hydrogen is low. <br> 4. Requirement of high light intensity. <br> 5. Oxygen is the limiting factor. |

**Table 6.** *Cont.*

| Mode of Operation | Advantages/Benefits | Disadvantages/Limitations |
|---|---|---|
| Indirect biophotolysis | 1. Hydrogen is produced from water in presence of sunlight.<br>2. $CO_2$ is decreased in the environment.<br>3. Pure hydrogen is obtained.<br>4. Nitrogenase enzymes can fix nitrogen and release a small amount of hydrogen simultaneously.<br>5. Separation of oxygen and hydrogen generation. | 1. $O_2$ inhibits the production of hydrogen.<br>2. Hydrogenase enzyme is sensitive to oxygen.<br>3. Productivity of hydrogen is low.<br>4. High energy is required.<br>5. Needs proper illumination. |
| Dark fermentation | 1. Various types of organic waste are used such as wastewater, sewage, farm waste, etc., as carbon sources.<br>2. Hydrogen production does not require light.<br>3. Hydrogen can be produced in the absence of light.<br>4. Waste from various sources can be managed; it moves toward more sustainable development.<br>5. Waste materials are converted into more valuable metabolites like butyric acid, lactic acid, and acetic acid.<br>6. Being an anaerobic process, it does not depend on oxygen. | 1. Hydrogen so obtained contains other gases, and purity is low.<br>2. Carbon and nitrogen are essential for the proper growth of microorganisms.<br>3. Separation of hydrogen from other gases is necessary.<br>4. The effluent of dark fermentation should be treated as it pollutes the environment because of the presence of undecomposed organic acids. |
| Photofermentation | 1. Various types of organic waste such as wastewater, sewage, farm waste, etc., as carbon sources.<br>2. Various spectrums or different light intensities can be used.<br>3. No oxygen generation, so it maintains an anaerobic environment.<br>4. Waste effluent of dark fermentation can be recycled.<br>5. Hydrogen produced is moderate with a high level of purity. | 1. Light is necessary for hydrogen production.<br>2. While constructing a reactor, proper alignment and exposure to the light are necessary.<br>3. Opaqueness in the effluent waste should be treated as it hampers the growth of microorganisms.<br>4. Oxygen is the limiting factor.<br>5. Solar energy is not fully utilized to form chemical energy. |
| Dark photofermentation | 1. Various types of organic waste such as wastewater, sewage, farm waste, the effluent of dark fermentation, VFAs, etc., as carbon sources.<br>2. It can occur in both conditions; that is, it can be performed in the presence or absence of light.<br>3. It produces hydrogen regardless of day–night cycles or light availability.<br>4. Microorganisms performing this process are more tolerant of oxygen.<br>5. It can yield relatively more amount of hydrogen. | 1. Overall energy conversion efficiency is low.<br>2. Because of the dark fermentative bacteria, hydrogen contains additional impure gases which may require an additional step for purification.<br>3. pH maintenance is a major concern as dark fermentation increases acidity.<br>4. Organism selection is a major concern in this case. Very specific microorganisms are needed with proper maintenance of environmental conditions. |

**Table 6.** *Cont.*

| Mode of Operation | Advantages/Benefits | Disadvantages/Limitations |
|---|---|---|
| Microbial electrolysis cell | 1. The highest hydrogen yield is seen.<br>2. Maintains efficient electron transfer between microorganisms and electrodes promoting organisms to produce hydrogen from organic substrate as an electron source.<br>3. Requires extremely low external potential for the microorganism to facilitate electron transfer.<br>4. Moves toward more sustainable energy development by treating wastewater and producing green energy.<br>5. A wide range of organic substrates such as wastewater, sewage, feedstocks, etc., can be used as a carbon source.<br>6. Removal of COD. | 1. High cost for construction and maintenance.<br>2. As there are many microorganisms present inside the reactors, maintenance of proper environmental conditions is always a challenge.<br>3. pH and temperature inside the reactor fluctuate frequently, and organisms are sensitive to the change.<br>4. Purity of gas is low, as it contains other gases. So additional purification step may be required.<br>5. Advanced technologies and designs are necessary for scaling up for commercial purposes. |

## 6. Limitations of Mathematical Modeling in Biohydrogen Production

Mathematical modeling helps in understanding and optimizing biohydrogen production processes. However, it also has limitations and challenges, especially when dealing with complex and dynamic biological systems. Some of the limitations of mathematical modeling in hydrogen production are as follows:

1. The mathematical models are based on simplified assumptions; hence, it may not fully capture the complexity of a real-time hydrogen production system, leading to inaccuracies.
2. Hydrogen production involves a series of chemical reactions, and each catalytic process has its own reaction kinetics, catalyst behavior, and rate-limiting factors and can follow multiple pathways. Modeling such complex reactions accurately is a challenge and may vary under different conditions.
3. Hydrogen production systems often exhibit nonlinear behavior. Therefore, a slight change in one of the parameters may lead to a significant change in the output.
4. The growth of one community or the addition of one nutrient substrate may inhibit the growth of another community; the modeling of these transient states accurately can be complex and may provide inaccuracy in hydrogen production predictions.
5. Several catalysts, both biological enzymes and chemical compounds, are involved, and catalyst deactivation or prediction of the longevity of the catalyst is challenging in mathematical modeling.
6. Mass balance transfer modeling is critical in many hydrogen production processes as these processes are complex and occur in multiphase and multi-physical systems.
7. Measurement error is very prominent in the biological world, so accurate modeling relying on high-quality data may not be readily available as it may be subjected to multiple errors in different steps.
8. Assessing the uncertainty and sensitivity of model predictions to parameter variations is important for robust modeling but can be computationally intensive.
9. Accuracy for validating mathematical models against experimental data is challenging as the experiments are costly and difficult to conduct on a large scale.
10. Scaling up from the laboratory scale to the industrial scale for hydrogen production is a challenge in mathematical modeling as it introduces additional complexities of equipment designs.

### 7. Way toward Future or Future Research Scope

1.  Only a few microbes are explored for biohydrogen production. There is always a way to explore and discover new microbial strains or use genetic engineering to create genetically modified microbes for higher hydrogen production.
2.  It is necessary to investigate the metabolic pathways of the microorganisms that are involved in hydrogen production and optimize the pathways by providing suitable nutrients or catalysts or making them favorable for better efficiency.
3.  It is also necessary to study the bioprocess optimization of parameters like temperature, pH, substrate concentration, and reactor design for developing advanced fermentation and bioprocessing techniques to maximize hydrogen production rate and yield.
4.  Developing effective catalysts for hydrogen evolution from different substrates remains a significant challenge in renewable hydrogen production. So, developing new catalysts to enhance hydrogen production opens a wide scope.
5.  It is necessary to design a scalable and cost-effective reactor design suitable for maintaining optimal growth conditions for hydrogen-producing microorganisms in industrial and laboratory conditions.
6.  It is necessary to develop more comprehensive mathematical models describing the kinetics of hydrogen-producing microorganisms with an account of parameters like substrate utilization, metabolic pathways, and extrinsic and intrinsic factors.
7.  Extensive research should be conducted to develop a mathematical model for the multiphase and multiscale modeling of mass transfer in biofilms, mixed populations of microbes, and reactor geometry.
8.  A proper mathematical model to predict the transient responses has not been formulated so far. A new study to develop dynamic models that can predict transient responses and adapt to changes in the environmental conditions of microbial populations can be a way forward.
9.  There is an open field of research on sensitivity and uncertainty analysis to assess the robustness of mathematical models' identification of key parameters affecting biohydrogen production.
10. The storage and transportation of hydrogen are still a problem. A wide field of research is open for modeling efforts to include the optimization of hydrogen storage and distribution systems in a cost-effective way.

### 8. Conclusions

This paper provides an up-to-date examination of mathematical modeling in various biological approaches to hydrogen production. The mathematical models and methods discussed encompass the entire spectrum of advancements, from their inception to the most recent state-of-the-art developments in hydrogen production using biological means. The paper includes a comparative analysis, identifies areas for future research, and highlights the limitations of previous studies. This comprehensive review will prove beneficial to researchers exploring diverse approaches to hydrogen production. It is important to note that the focus of this review is primarily on biological methods, such as thermal production, electrolysis, thermolysis, and renewable and photoelectrochemical methods. While these methods are covered extensively, it should be acknowledged that this list is not exhaustive, and there are other approaches to hydrogen production not discussed in this paper.

**Author Contributions:** Conceptualization, P.D.; methodology, P.Y. and P.D.; software, P.Y.; validation, P.Y. and P.D.; formal analysis, P.Y.; investigation, P.Y.; resources, P.Y. and P.D.; data curation, P.Y. and P.D.; writing—original draft preparation, P.Y.; writing—review and editing, P.Y. and P.D.; visualization, P.Y.; supervision, P.D.; project administration, P.D. All authors have read and agreed to the published version of the manuscript.

**Funding:** This research received no external funding.

**Data Availability Statement:** The data presented in this study are openly available in article.

**Conflicts of Interest:** The authors declare that they have no known competing financial interest or personal relationships that could have appeared to influence the work reported in this paper.

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
