# Peer review of "A Review on Mathematical Modeling of Different Biological Methods of Hydrogen Production"

_hydrogen, doi:10.3390/hydrogen4040053_

Round 1
Reviewer 1 Report
Comments and Suggestions for Authors
Dear authors,
the manuscript provides an extensive review of biological processes to produce hydrogen, focusing on modelling and available experimental data.
However, in the actual form the manuscript is suggested to be published only after major revions.
First of all all, it is recommended to change the title which is misleading, since the review focuses only on biological methods for hydrogen production.
The short overview of non-biological methods in the introduction is fine and necessary. However, I would ask to rework the structure of the presented processes along with a check, that in each case comparable information is given. It is suggested to rework Fgure 1 and then follow strictly the structure given there: For example I see coal gasification as a thermal production process. Electrolysis based on photovoltaics is just another sub-process of electrolysis.
Finally, it is recommended to rework and restructure the description of the different biological processes:
- I see Eq. 3 as overall reaction of Eq. 4 and 5. At is a t this point not clear where Eq. 6-7 are coming from. Why is nitrogen playing a role there. This needs to be explained in a proper introduction
- Given equations needs more explanations: What is the genearal basis for modeling of a specific process: Monod, Gompertz, etc.? What are the main influencing parameter - enumerate in short at the beginn of the mathematical description. Which are optional equations (e.g. Line 221 and 223, Line 740 and 148) and when to use them?
- In general I suggest to start with the equation of hydrogen evolution, because it is the aim of work (eg. Line 753 and 755 at the very end!). Based on that the calculation of of the parameters should be explained.
- Is it possible to convert units in table 1 to compare the results of given references?
- Table 3 summarizing dark and photofermentation is quiet short. Furthermore, use of kinetic and parametric models is not quiet clear. To which kind of model the equations given before in this chapter belong?
- In chapter 4 besides the simplified biofim model and the time dependent growth model, also a production time space model is mentioned, but not further described.
What's also besides the advantages and disadvantages of processes some kind of conclusion is missing concerning modeling: Are the models sufficient and all effects covered? Which models to prefere? How do models fit the the experimental data? What are recommendation for further research?
I am looking forward to receive a reworked version of the manuscript!
Comments on the Quality of English LanguagePleas check in general language and readability!
Author Response
Respected reviewer, the revision has been done as per your suggestions. Please see the attached file for details.

Reviewer 2 Report
Comments and Suggestions for Authors
The submitted manuscript falls within the hydrogen research scope; however, it requires significant revisions before it could be considered for publication. My specific comments are as follows:
- The title and abstract should include definitions of the various methods being discussed. This will help readers understand the focus of the topic, as the manuscript does not cover all hydrogen production methods.
- The abstract needs to be longer and needs to be expanded. The authors should highlight the originality of their study and specify which hydrogen production methods are the focus of their review paper.
- The introduction requires expansion. Since hydrogen production is a popular area of research with extensive literature, the authors should cite more relevant papers. This will serve as a basis for explaining the novelty and originality of their work.
- The presentation of mathematical models for different hydrogen production methods needs improvement. All limitations, parameters, and assumptions should be clearly stated.
- The authors may also consider incorporating mathematical modeling and simulation techniques for water electrolyzer systems integrated with renewable energy sources, which could enhance the paper's impact.
- The authors should include several figures related to the discussed topics to aid in reader comprehension.
- A concluding remarks section should be added to summarize the key points and contributions of the review paper.
- Future directions for research should also be discussed to provide context for ongoing or upcoming studies in the field.
Given these considerations, the manuscript's current version is unsuitable for publication. I recommend the authors carefully revise this version and resubmit it to the journal for further evaluation.
Author Response

(The authors gave the same response as above.)

Reviewer 3 Report
Comments and Suggestions for Authors
In the manuscript " A Review on Mathematical Modelling of Different Methods of Hydrogen Production" Pradip Debnath et al. presents an updated review on mathematical modelling of different methods of hydrogen production. The presented mathematical modelling and methods range from inception to the current state of the art developments in hydrogen production. A comparative study has been done along with indication for future research and short comings of earlier research. This manuscript is well-organized. It can be accepted after minor revision. The comments are presented as follows:
1. The latest literature about hydrogen production technologies should be cited, such as Metal organic framework supported niobium pentoxide nanoparticles with exceptional catalytic effect on hydrogen storage behavior of MgH2. L. Zhang, F.M. Nyahuma, H. Zhang, C. Cheng, J. Zheng, F. Wu, L. Chen. Green Energy and Environment, 2022, https://doi.org/10.1016/j.gee.2021.09.004. Chao Wan, Liu Zhou, Suman Xu, Biyu Jin, Xin Ge, Xing Qian, Lixin Xu, Fengqiu Chen, Xiaoli Zhan, Yongrong Yang, Dangguo Cheng. Defect engineered mesoporous graphitic carbon nitride modified with AgPd nanoparticles for enhanced photocatalytic hydrogen evolution from formic acid, Chemical Engineering Journal, 2022, 429, 132388. Wang Yaxiong, Zhong Shunbin, Sun Fengchun. Research Progress in Vehicular High Mass Density Solid Hydrogen Storage Materials. Chinese Journal of Rare Metals, 2022, 46, 796-812.
2. What is the main question addressed by the research?
3. What does it add to the subject area compared with other published material?
4. Do you consider the topic original or relevant in the field?
Author Response

(The authors gave the same response as above.)

Round 2
Reviewer 1 Report
Comments and Suggestions for Authors
Dear authors,
thank you for reworking the manuscript. The title now fits the content of the paper. Also added sections 6 and 7 improve the quality of the paper.
However, the recommendation recommended to rework and restructure the description of the different (biological) processes is followed just partially. Thus, before publication of the manuscript further major revisions.
Finally, it is recommended to rework and restructure the description of the different biological processes:
- There are still some short-comings in Figure 1: No respective methods are defined for “thermochemical or thermolysis”. “Renewable” is not a specific method but is possible for “biohydrogen” and “electrolysis systems” – “photvoltaic electrolysis systems” are specifically named as a sub-method.
- Introduction: Hydrogen is not only seen as energy carrier but more and more as important component for synthesis. Try to describe methods in the introduction following the structure given in Figure 1! Take care that each method presented in Figure 1 is also covered in the introduction.
- Biophotolysis: Improve the general description – In line 95 it is stated, that volatile organic substances are used in contrast to line 98 where CO2 is defined as carbon source. What’s “Fd” in equations 1 and 2 – mention in text! Equation 5 can not be the overall reaction, but the first (aerobic ) step of indirect biophotolysis. Where is equation 6 happening describe and refer in the text!
- Given model equations: adjust spacing – lines 177-179 belongs to line 175 (equation 9) Then an empty line is suggested since reference Vargas, 2016 seems to belong to equations 10 and 11, together with line 186. Afterwards again an empty line etc.
- Line 202, equation 16 – does this equation belong to algae or cyanobacterial growth – or both?
- Equations 19 and 21: which specific growth rate is used for which case/condition?
- Line 265: unit “nmol/mg chl(?) a(ß)/h”?
- Equation 60 and 61 have the same structure and can be combined.
- Chapter 5: Never ever start a chapter with a table or Figure. At least some introduction needs to be given. The text given in lines 1041-1060 does not fit the chapter title! If necessary it could be shifted to the end of introduction section of the paper to cover the aspect of storage – however, the title of the review is “ .. Hydrogen production …”. Why not combining chapters 5 and 6?
I am looking forward to receive a reworked version of the manuscript!
Author Response
We are very thankful to the respected reviewer. The manuscript has been revised as per comments. Please check the attached file for details.

Reviewer 2 Report
Comments and Suggestions for Authors
The authors appropriately revised the previous version of the manuscript.
Author Response
We are very thankful to the respected reviewer for appreciating the revised version. Some new changes have been made to the paper. Please see the attached file for details.

Round 3
Reviewer 1 Report
Comments and Suggestions for Authors
No further comments and suggestions! Thank you for your corrections!